# Contrastive Learning with Consistent Representations

**Zihu Wang**                                                              *zihu_wang@ucsb.edu*
*Department of Electrical and Computer Engineering*
*University of California, Santa Barbara*

**Yu Wang**                                                                    *yu95@ucsb.edu*
*Department of Electrical and Computer Engineering*
*University of California, Santa Barbara*

**Zhuotong Chen**                                                            *ztchen@ucsb.edu*
*Department of Electrical and Computer Engineering*
*University of California, Santa Barbara*

**Hanbin Hu**                                                              *hanbinhu@ucsb.edu*
*Department of Electrical and Computer Engineering*
*University of California, Santa Barbara*

**Peng Li**                                                                      *lip@ucsb.edu*
*Department of Electrical and Computer Engineering*
*University of California, Santa Barbara*

**Reviewed on OpenReview:** *https://openreview.net/forum?id=gKeSI8w63Z*

## Abstract

Contrastive learning demonstrates great promise for representation learning. Data augmentations play a critical role in contrastive learning by providing informative views of the data without necessitating explicit labels. Nonetheless, the efficacy of current methodologies heavily hinges on the quality of employed data augmentation (DA) functions, often chosen manually from a limited set of options. While exploiting diverse data augmentations is appealing, the complexities inherent in both DAs and representation learning can lead to performance deterioration. Addressing this challenge and facilitating the systematic incorporation of diverse data augmentations, this paper proposes Contrastive Learning with Consistent Representations (`CoCor`). At the heart of `CoCor` is a novel consistency metric termed DA consistency. This metric governs the mapping of augmented input data to the representation space. Moreover, we propose to learn the optimal mapping locations as a function of DA. Experimental results demonstrate that `CoCor` notably enhances the generalizability and transferability of learned representations in comparison to baseline methods. The implementation of `CoCor` can be found at `https://github.com/zihuwang97/CoCor`.

## 1 Introduction

Data augmentation (DA) is widely used in supervised learning in computer vision Ho et al. (2019); Lim et al. (2019); Cubuk et al. (2019; 2020); Li & Li (2023), achieving excellent results on popular datasets Ciregan et al. (2012); Sato et al. (2015); Wan et al. (2013); Krizhevsky et al. (2017). DA is also a key component in recent contrastive learning techniques Chen et al. (2020a); Tian et al. (2020b); He et al. (2020); Chen & He (2021); Xiao et al. (2020); Lee & Shin (2023). An encoder that learns good visual representations of the input data is trained with a contrastive loss. The contrastive loss is characterized by the following principle: in the feature space, two views of a given data example, transformed by distinct DA functions, exhibit correlation (similarity), whereas transformed views of different input examples manifest dissimilarity. The effectiveness

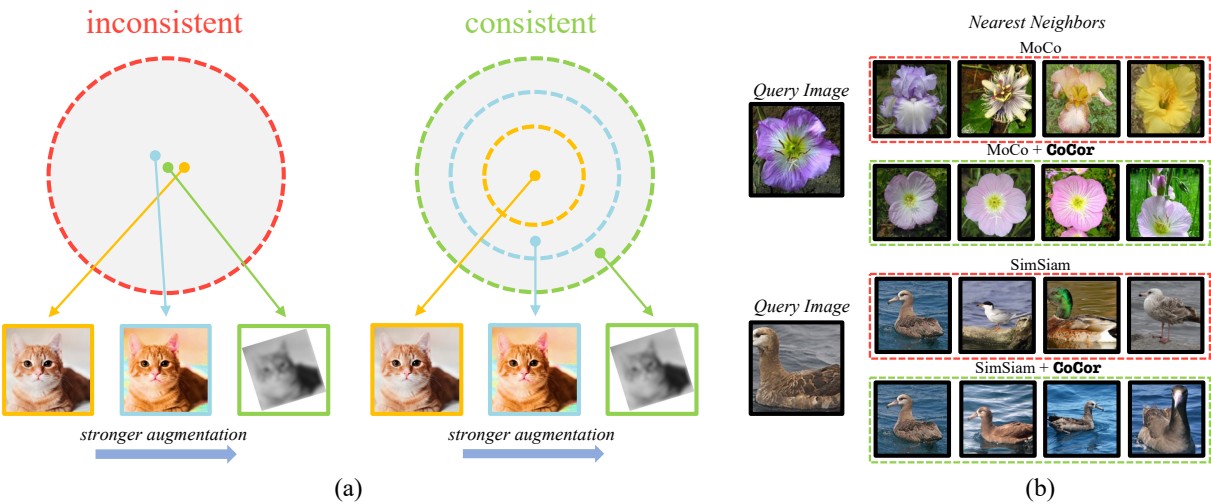

Figure 1: (a) Left: An encoder trained with the standard contrastive loss can exhibit inconsistency, as different views of an instance are encouraged to be represented similarly in the feature space, irrespective of the actual difference between them. Right: A consistent encoder positions the vector of more strongly augmented data further away from that of the raw data. Here the rings represent points with varying similarities to the central representation vector. (b) Nearest-neighbor retrieval in the feature space on CUB-200 Wah et al. (2011) and Flowers102 Nilsback & Zisserman (2008) using pre-trained encoders. Existing contrastive methods He et al. (2020); Chen & He (2021), which enforce invariance to all data augmentations, may inadvertently cluster dissimilar data closely in the feature space. However, by applying *consistency*, CoCor ensures that only data sharing similar latent semantics are distributed closely in the latent space.

of the encoder, trained on unlabeled data, is pivotal to the overall performance of contrastive learning and is contingent upon the choice of employed DAs.

To learn effective and transferable representations, numerous studies have focused on enhancing contrastive learning by selecting suitable DAs. Chen et al. (2020a); Tian et al. (2020b); He et al. (2020); Li et al. (2023); Wang et al. (2023); Van der Sluijs et al. (2024) have shown that combining different augmentations with appropriate intensities can improve performance. However, these works confine augmentations to a random composition of specific types within a limited intensity range. Contrastingly, research has indicated that employing diverse DAs effectively enhances the model's ability to capture the invariance of the training data, thereby boosting the model's performance Cubuk et al. (2020), transferability Lee et al. (2021), and robustness Lopes et al. (2019). Consequently, the exploration of diverse DAs has garnered increased attention recently. PIRL Misra & Maaten (2020) adopts additional augmentations, SwAV Caron et al. (2020) introduces multiple random resized-crops to provide the encoder with a broader range of data reviews. Moreover, some recent works adopt combinations of stronger DAs in contrastive learning Tian et al. (2020b); Wang & Qi (2021); Lee & Shin (2023).

While the idea of leveraging diverse data augmentations is appealing, to date, there lacks a systematic approach for integrating a substantial number of augmentations into contrastive learning. It is crucial to note that the intricacies of data augmentation and representation learning may lead to a performance degradation if augmentation functions are not judiciously chosen Tian et al. (2020b); Chen et al. (2020a); Wang & Qi (2021).

To tackle the aforementioned challenges, this paper introduces Contrastive Learning with Consistent Representations (CoCor). First, we define a set of composite augmentations, denoted as $\mathbf{\Omega_c}$, formed by combining multiple basic augmentations. Each composite augmentation is represented using a composition vector that encodes the types and frequencies of the basic augmentations employed. This composite set $\mathbf{\Omega_c}$ facilitates the incorporation of diverse DAs, encompassing a wide range of transformation types and overall intensity.

Crucially, we introduce the concept of consistency for data augmentations, termed *DA consistency*, which imposes a desired property on the representations of views generated by various composite data augmentations. Specifically, for a given input example, a stronger data augmentation should move its feature space view farther from the representation of the original example. In other words, the similarity in feature space between the transformed view and the original input decreases with the strength of the data augmentation, as depicted in Figure 1(a). An encoder failing to satisfy this property is considered inconsistent from a data augmentation perspective.

We propose the integration of a novel *consistency loss* to enforce *DA consistency* within the representation space of the encoder. Unlike contrastive loss, which does not account for the type or strength of the data augmentations used Chen et al. (2020a); Tian et al. (2020b); He et al. (2020), our *consistency loss* strategically guides encoder training to ensure alignment of the resultant feature space with the strength of the applied data augmentations. As demonstrated in nearest-neighbor retrieval tasks shown in Figure 1(b), existing contrastive learning approaches like He et al. (2020); Chen et al. (2020b); Chen & He (2021) may cluster dissimilar views due to their agnostic to augmentation-induced data variance. However, the consistency enforced by `CoCor` ensures proximity in the feature space only for data with similar latent features, while distinctly separating variants of the data. Furthermore, our experimental studies affirm that maintaining *DA consistency* is crucial for mitigating discrepancies among diverse data augmentations, thereby preventing potential performance degradation or even dimensional collapse in the feature space Jing et al. (2021); Li et al. (2022a).

The proposed consistency loss hinges on determining the optimal similarity between the representations of an input example and its transformed view, a value contingent on the strength of the composite data augmentation. However, this optimal similarity is not known *a priori*. To address this challenge, we adopt a data-driven approach, training a neural network to map from the composition vector of each data augmentation to the desired similarity. Recognizing the monotonic relationship between similarity and augmentation strength, we introduce and enforce a monotonicity constraint on the neural network. This constraint ensures that stronger composite data augmentations correspond to a strictly smaller valued similarity.

The proposed Contrastive Learning with Consistent Representations (`CoCor`) has the following contributions:

- Systematically explore a large set of diverse composite data augmentations to improve the encoder's performance on various downstream tasks.

- Propose a novel concept, *DA consistency*, which quantifies the monotonic dependency of latent-space similarity between an input example and its transformed view on the strength of the augmentation.

- Introduce a new *consistency loss* designed to train the encoder to satisfy *DA consistency* and to utilize diverse augmentations without suffering inconsistency-induced performance loss.

- Train a monotonically constrained neural network for learning the optimal latent-space similarities.

With consistent representations, `CoCor` achieves state-of-the-art results for various downstream tasks. Moreover, it can be readily integrated into existing contrastive learning frameworks, effectively imposing DA consistency on the encoder.

## 2 Related Work

**Contrastive Learning** demonstrates significant potential in computer vision tasks and other domains in recent years Chen et al. (2020a); He et al. (2020); Wang et al. (2023); Caron et al. (2020); Wang et al. (2024). It is a common practice to utilize data augmentations in forming both positive and negative pairs of data for defining the contrastive loss Chen et al. (2020a); Oord et al. (2018). Some methods highlight the significance of negative pairs in learning distinguishable features from the data Robinson et al. (2020); Awasthi et al. (2022). MoCo introduces a moving-averaged encoder paired with a large negative memory queue He et al. (2020); Misra & Maaten (2020). In contrast, Chen & He (2021); Grill et al. (2020) exclusively learn features from positive pairs, achieving state-of-the-art performance without incorporating negative

pairs. Furthermore, supervised contrastive methods introduce annotation-related information to learn better representations Khosla et al. (2020); Li et al. (2022b). Recent research endeavors aim to comprehend the effectiveness and limitations of contrastive learning via the lens of representation distribution Wang & Isola (2020) and delve into issues such as dimensional collapse Li et al. (2022a); Jing et al. (2021).

**Data Augmentation** Data augmentations applied to natural images have demonstrated efficacy in enhancing the generalizability and robustness of models trained in supervised learning Cubuk et al. (2019; 2020); Lopes et al. (2019); Krizhevsky et al. (2017). However, in self-supervised learning Chen et al. (2020a); He et al. (2020), careful selection of data augmentations from a limited set becomes crucial for ensuring optimal performance. Instead of treating both views in positive pairs equally, recent proposals by Zhang & Ma (2022); Lee et al. (2021); Zhang et al. (2022); Devillers & Lefort (2023) suggest capturing the variance between two views caused by the application of different random augmentations, termed augmentation-aware information. AugSelf Lee et al. (2021) achieves this by training an auxiliary network to predict the difference between augmentations applied to generate positive pairs. LoGo Zhang et al. (2022) proposes learning the differences between variously sized crops of each image. Zhang & Ma (2022) encodes an augmented image along with the DA parameters using two networks, and combines the two embeddings to form a feature vector for contrastive loss. EquiMod Devillers & Lefort (2023) trains a network to predict the representation of an augmented view from the original data and the applied DA. While methods like InfoMin Tian et al. (2020b) and JOAO You et al. (2021) propose searching for optimal data augmentation policies for forming pairs in contrastive loss, their empirical studies reveal that stronger and more diverse augmentations can, in fact, degrade encoder performance. Conversely, some recent works aim to explore the benefits of strong augmentations Wang & Qi (2021); Caron et al. (2020); Lee & Shin (2023). However, as of now, there is a lack of a systematic approach to leveraging diverse data augmentations while considering the impact of augmentation strength.

## 3 Preliminaries

The goal of contrastive representation learning is to learn an encoder parameterized by $\boldsymbol{\theta_e}$ that maps an input data $\boldsymbol{x} \in \mathbb{R}^n$ to an $\ell_2$ normalized feature vector $\boldsymbol{z}$ of dimension $m$ in the feature space $\mathcal{Z}$, i.e., $f_{\boldsymbol{\theta_e}}(\cdot) : \mathbb{R}^n \rightarrow \mathcal{S}^{m-1}$. The encoder is trained by minimizing a contrastive loss. Specifically, it defines two different augmented versions of the same input example as a positive pair, which are expected to have similar representations in the feature space. Meanwhile, the encoder shall be trained to discriminate any two instances augmented from different input examples, i.e., a negative pair, in the feature space. Minimizing the contrastive loss pulls together positive pairs and pushes apart negative pairs Oord et al. (2018); Chen et al. (2020a); Li et al. (2020).

## 4 Method

### 4.1 Diverse Composite Augmentations

The proposed Contrastive Learning with Consistent Representations (`CoCor`) improves contrastive learning performance by exploring diverse data augmentations composed from a set of basic augmentations.

**Definition 1 (Composite Data Augmentations).** A composite augmentation with a length of $l$, namely, $A_i^{<l>}$, is defined as the composition of $l$ randomly sampled augmentations from a given set of $N_a$ basic augmentation functions $\boldsymbol{\Omega}_a = \{a_1(\cdot), a_2(\cdot), \cdots, a_{N_a}(\cdot)\}$: $A_i^{<l>} = a_{i(1)} \circ a_{i(2)} \circ \cdots a_{i(l)}$, where $i(k) \in [1, N_a], \ k = 1, 2 \cdots l$.

We denote the set of all composite augmentations with a length of $l$ by $\boldsymbol{\Omega}_c^{<l>}$, and denote the set of all composite augmentations with different lengths by $\boldsymbol{\Omega}_c = \boldsymbol{\Omega}_c^{<1>} \cup \boldsymbol{\Omega}_c^{<2>} \cdots$. Our studies show that the ordering of the basic augmentations in a composite augmentation does not have a significant impact on performance. To simplify the discussion, we assume that composite augmentations are order invariant, e.g., $a_i \circ a_j = a_j \circ a_i$. We represent a composite augmentation with an unique composition vector as shown in Figure 2(b).

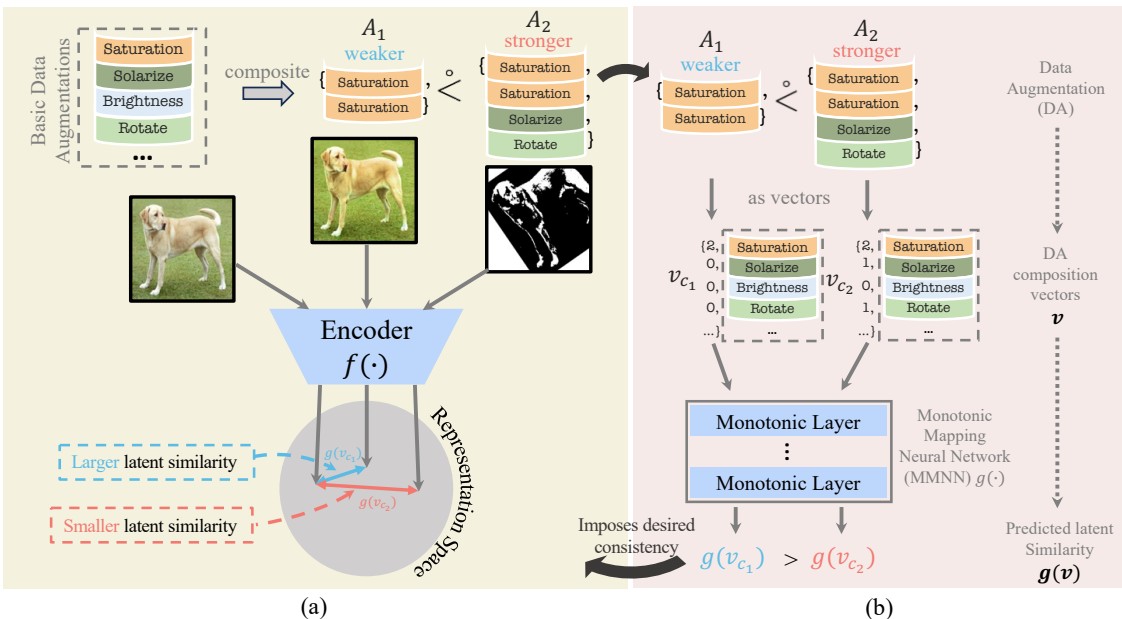

Figure 2: Overview of the Proposed Method. (a) The proposed property, *DA consistency*, ensures that data augmented with stronger augmentation is positioned farther from the original data compared to a weakly augmented view in the representation space. (b) The Monotonic Mapping Neural Network (MMNN) predicts the optimal latent similarity between an augmented view and the original data, using augmentation composition vectors as input. Stronger augmentation results in a smaller predicted latent similarity by the MMNN.

**Definition 2** (**Composition Vector**). The composition vector of a length-$l$ composite augmentation $A_i^{<l>}$ is defined as a $N_a$-dimensional vector $\boldsymbol{v}_i = \boldsymbol{v}(A_i^{<l>}) \in \mathbb{N}_0^{N_a}$ ($\mathbb{N}_0$ is the set of all natural numbers), where the $j_{th}$ entry $\boldsymbol{v}_i[j]$ is the number of times that the basic augmentation $a_j$ is applied in $A_i^{<l>}$, and $\sum_{j=1}^{N_a} \boldsymbol{v}_i[j] = l$.

## 4.2 Data Augmentation Consistency

Data augmentation plays a crucial role in contrastive learning, requiring careful design to expose the latent structure of the original data. However, our key observation is that the standard contrastive learning framework Chen et al. (2020a); He et al. (2020); Chen & He (2021) does not inherently imply *data augmentation (DA) consistency*, a fundamental property proposed by this work. In the absence of this consistency, the encoder might simply bring views in positive pairs together in the feature space $\mathcal{Z} \subseteq \mathcal{S}^{m-1}$, irrespective of the actual difference between them. However, a weakly augmented view and a strongly augmented view of an input example may not necessarily share the same latent structure, and it could be advantageous to encode them into different regions in the feature space, as illustrated in Figure 1 (a). In such cases, an encoder attempting to pull positively paired views together would learn an incorrect representation distribution.

To this end, we introduce the concept of *DA consistency*. With the *DA consistency* imposed, the encoder is trained not only to cluster positive pairs together but also to map them to their respective optimal locations in the latent feature space $\mathcal{Z}$ based on the strength and types of applied augmentations. Consequently, the encoder preserves critical information introduced by data augmentations, such as variances related to brightness, sharpness, and rotation, as shown in Figure 2(a). This characteristic of the pre-trained encoder enhances its performance on various downstream tasks, where such information is crucial for recognition.

We quantify a mapped feature space location using *latent similarity* with respect to the raw input data.

**Definition 3** (**Latent Similarity**)**.** The latent similarity $l_d(\boldsymbol{x}; f, A)$ of an input $\boldsymbol{x}$, given an encoder $f$ and a composite augmentation $A$, is defined as the cosine similarity between the normalized representations of $\boldsymbol{x}$ and $A(\boldsymbol{x})$ in the feature space $\mathcal{Z} \subseteq \mathcal{S}^{m-1}$: $l_d(\boldsymbol{x}; f, A) = f(\boldsymbol{x})^T \cdot f(A(\boldsymbol{x}))$.

To be DA consistent, latent-space similarities of different augmented views of the same data shall be monotonically decreasing with the augmentation strength. We learn a mapping, i.e., a parameterized neural network $g_{\boldsymbol{\theta_d}}(\cdot)$ that takes a composition vector $\boldsymbol{v}(A)$ of a composite augmentation $A$ as input and map it to the optimal latent similarity $l_d^*(\boldsymbol{v}(A))$, as detailed in Section 4.3. An optimal encoder is considered to be fully DA consistent, if its latent similarity for any given composite augmentation is considered optimal. For encoders that are not fully DA consistent, we further define DA consistency level, as the degree at which the learned representations are consistent in terms of data augmentation.

**Definition 4** (**Consistency Level**)**.** Given an encoder $f$, a set of composite augmentations $\boldsymbol{\Omega}_c$, and a raw input $\boldsymbol{x}$, the DA consistency level (DACL) is define as the encoder's deviation from optimal latent similarity:

$$\text{DACL}(\boldsymbol{x}; f, \boldsymbol{\Omega}_c) = \mathbb{E}_{A \sim \boldsymbol{\Omega}_c}[|l_d(\boldsymbol{x}; f, A) - l_d^*(\boldsymbol{v}(A))|] \tag{1}$$

DACL measures the level of consistency of a given encoder $f$ by comparing the latent similarity of the augmented view of $\boldsymbol{x}$ with the corresponding optimal latent similarity $l_d^*(\boldsymbol{v}(A))$ over all possible composite augmentations.

With DACL, an encoder $f_{\boldsymbol{\theta_e}}(\cdot)$ parameterized by $\boldsymbol{\theta_e}$ that is not fully DA consistent can be trained to be more DA consistent by minimizing the *consistency loss* defined below:

$$\begin{aligned}
\mathcal{L}_{\text{consistent}}(\boldsymbol{\theta_e}) &= \mathbb{E}_{\boldsymbol{x} \sim \mathcal{X}, A \sim \boldsymbol{\Omega}_c}[|\text{DACL}(\boldsymbol{x}; f_{\boldsymbol{\theta_e}}, \boldsymbol{\Omega}_c)|] \\
&= \mathbb{E}_{\boldsymbol{x} \sim \mathcal{X}, A \sim \boldsymbol{\Omega}_c}[|l_d(\boldsymbol{x}; f_{\boldsymbol{\theta_e}}, A) - l_d^*(\boldsymbol{v}(A))|] \\
&= \mathbb{E}_{\boldsymbol{x} \sim \mathcal{X}, A \sim \boldsymbol{\Omega}_c}[|l_d(\boldsymbol{x}; f_{\boldsymbol{\theta_e}}, A) - g_{\boldsymbol{\theta_d}}(\boldsymbol{v}(A))|]
\end{aligned} \tag{2}$$

Our empirical study shows that an alternative consistency loss provides better performance, details of this alternative is included in Appendices.

### 4.3 Learning Optimal Latent Similarities

#### 4.3.1 Learnable Model for Optimal Latent Similarities

To train the encoder on the proposed consistency loss in equation 2, we need to provide the optimal latent similarity $l_d^*(\boldsymbol{v}(A))$ for a given composite augmentation $A$, which is contingent on the strength of $A$. Nevertheless, defining augmentation strength poses a challenge, as two composite augmentations may comprise different basic augmentation types, making it unclear how to define and compare their strength.

To tackle this problem, we learn a mapping, i.e., a parameterized neural network $g_{\boldsymbol{\theta_d}}(\cdot)$ that takes a composition vector as input and map it to the optimal latent similarity $l_d^*(\boldsymbol{v}(A))$. This approach relaxes the need for directly modeling the strength of different augmentations.

#### 4.3.2 Imposing Monotonicity to the Learnable Model

Taking it a step further, we not only approximate the optimal latent similarity $l_d^*$ with a neural network but also enforce an important monotonic property to better learn the desired optimal latent similarities.

As the overall strength of a composite augmentation increases, e.g., by incorporating additional basic augmentations, the similarity between the augmented data and the raw data is expected to decrease. For instance, although directly comparing the effects of two different augmentations `GaussianBlur` and `Sharpness` (which blurs the image by a Gaussian kernel and adjusts edge contrast, respectively.) may be challenging, applying `Sharpness` twice in a composite DA would certainly distort the raw input more than applying it once.

**[DA Strength Comparison Operator]**Thus, as illustrated in Figure 3, formally, we define an operator $\overset{\circ}{>}$ for comparing the strength of composition data augmentations in set $\mathbf{\Omega}_c$: $A_i \overset{\circ}{>} A_j$ iff $\boldsymbol{v}(A_i) \overset{\circ}{>} \boldsymbol{v}(A_j)$, where $\overset{\circ}{>}$ operates element wise on the composition vectors: $\boldsymbol{v}(A_i) \overset{\circ}{>} \boldsymbol{v}(A_j)$ implies that $\boldsymbol{v}(A_i)[k] \geq \boldsymbol{v}(A_j)[k], \forall k \in [1, N_a]$, and $\exists k \in [1, N_a]$ such that $\boldsymbol{v}(A_i)[k] > \boldsymbol{v}(A_j)[k]$.

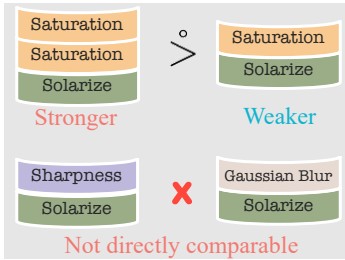

In other words, composite augmentation $A_i$ is considered to be stronger than $A_j$ (i.e., $A_i \overset{\circ}{>} A_j$) if and only if the number of times any basic augmentation is applied in $A_i$ is at least that of the same basic augmentation is applied in $A_j$, and there is at least one basic augmentation used in $A_i$ more times than in $A_j$.

Figure 3: A composite augmentation is considered stronger than another if and only if it includes all components of the latter, along with additional basic augmentations.

Our learnable neural network model $g_{\boldsymbol{\theta}_d}(\cdot)$ takes the composition vector of an augmentation $A_i$ as the input, and map it to the optimal latent similarity of $l_d^*(\boldsymbol{v}(A_i))$ in the feature space: $l_d^*(\boldsymbol{v}(A_i)) = g_{\boldsymbol{\theta}_d}(\boldsymbol{v}(A_i))$. We enforce monotonicity w.r.t. input $\boldsymbol{v}(A_i)$ on $g_{\boldsymbol{\theta}_d}(\cdot)$: $g_{\boldsymbol{\theta}_d}(\boldsymbol{v}(A_i)) > g_{\boldsymbol{\theta}_d}(\boldsymbol{v}(A_j))$ if $v(A_i) \overset{\circ}{>} v(A_j)$, $\forall A_i, \forall A_j \in \mathbf{\Omega}_c$. In other words, **stronger data augmentations result in a smaller latent similarity**. The monotonicity is enforced by incorporating monotonic linear embedding layers You et al. (2017) in $g_{\boldsymbol{\theta}_d}(\cdot)$, as shown in Figure 2(b). We refer to $g_{\boldsymbol{\theta}_d}(\cdot)$ as a monotonic mapping neural network (MMNN).

## 4.4 Encoder Training with Consistency

The proposed `CoCor` approach incorporates the consistency loss in equation 2 to form the overall loss function $\mathcal{L}_u$ for optimizing the encoder parameter $\boldsymbol{\theta_e}$ given the MMNN's parameter $\boldsymbol{\theta_d}$ on unlabeled data:

$$\mathcal{L}_u(\boldsymbol{\theta_e}|\boldsymbol{\theta_d}) = \mathcal{L}_{\text{contrast}}(\boldsymbol{\theta_e}) + \mathcal{L}_{\text{consistent}}(\boldsymbol{\theta_e}|\boldsymbol{\theta_d}) \tag{3}$$

In equation 3, the optimization of the encoder parameters, $\boldsymbol{\theta_e}$, is contingent upon the MMNN parameters, $\boldsymbol{\theta_d}$. Consequently, we denote the dependency of $\boldsymbol{\theta_e}$ on $\boldsymbol{\theta_d}$ as $\boldsymbol{\theta_e}(\boldsymbol{\theta_d})$ and solve a bi-level optimization problem to optimize $\boldsymbol{\theta_d}$:

$$\min_{\boldsymbol{\theta_d}} \quad \text{CE}(\boldsymbol{\theta_e^*}(\boldsymbol{\theta_d})) \tag{4}$$

$$\text{s.t.} \quad \boldsymbol{\theta_e^*}(\boldsymbol{\theta_d}) = \arg\min_{\boldsymbol{\theta_e}} \mathcal{L}_u(\boldsymbol{\theta_e}|\boldsymbol{\theta_d}) \tag{5}$$

The top-level problem in equation 4 optimizes the MMNN and a linear classifier (not explicitly shown in equation 4 based on a cross entropy loss CE on a small amount of labeled data. During each iteration, the bottom-level problem in equation 5 is solved to update the encoder parameter $\boldsymbol{\theta_e}$ on the unlabeled data. We defer the derivation of $\theta_e$'s dependency on $\theta_d$ and update rule for $\theta_d$ to Appendices.

We conclude the algorithm flow of the proposed approach in Algorithm 1.

The design of `CoCor` makes it compatible with various state-of-the-art self-supervised learning frameworks. `CoCor` can be integrated into an existing method by replacing $\mathcal{L}_{\text{contrast}}$ in equation 3 with the method's specific loss function. The added consistency loss imposes DA consistency and improves the generalizability and transferability of the pre-trained encoder within these frameworks.

---

**Algorithm 1:** Algorithm flow of `CoCor`

---

**Input:** initial encoder, MMNN, and classifier parameters $\boldsymbol{\theta_e^{(0)}}$, $\boldsymbol{\theta_d^{(0)}}$, and $\boldsymbol{\theta_c^{(0)}}$, number of training epochs $N$, unlabeled dataloader $\mathcal{D}_u$, labeled dataloader $\mathcal{D}_l$.

**for** $i=1$ **to** $N$ **do**

    1.Sample unlabeled $\boldsymbol{x}_u$ and labeled data $(\boldsymbol{x}_l, \boldsymbol{y}_l)$ from $\mathcal{D}_u$ and $\mathcal{D}_l$, respectively.

    2.Call equation 5 to update the encoder's parameter $\boldsymbol{\theta_e^{(i)}}$ on $\boldsymbol{x}_u$ with MMNN parameters $\boldsymbol{\theta_d^{(i-1)}}$ fixed.

    3.Call equation 4 to update MMNN $\boldsymbol{\theta_d^{(i)}}$ and classifier $\boldsymbol{\theta_c^{(i)}}$ on $(\boldsymbol{x}_l, \boldsymbol{y}_l)$ with the encoder parameters $\boldsymbol{\theta_e^{(i)}}$ fixed.

**Output:** Trained encoder with parameters $\boldsymbol{\theta_e^N}$

---

## 5 Experimental Studies

We demonstrate the performance and generality of `CoCor` by integrating it into several state-of-the-art contrastive learning methods: MoCo Chen et al. (2020b); He et al. (2020), a dual-encoder approach with a large negative memory queue, SimSiam Chen & He (2021), which exclusively leverages positive pairs, and SupCon Khosla et al. (2020), a supervised contrastive learning method that uses labeled data to enhance representation learning. We refer to each of the three methods as a baseline. We compare the performance of encoders pre-trained using these baselines against those pre-trained using a combination of `CoCor` with the baselines across a variety of datasets for linear evaluation and object detection. `CoCor` is also compared with recent works which take augmentation-aware information into consideration Zhang & Ma (2022); Lee et al. (2021); Devillers & Lefort (2023) and works which use stronger data augmentations than conventional contrastive learning Lee & Shin (2023); Wang & Qi (2021). To shed more light on the working mechanism of `CoCor`, we perform post-hoc analysis on latent similarity and potential dimensional collapse, and a number of ablation studies.

Unless otherwise noted, all experimental results presented in this paper are reproduced by us.

### 5.1 Experimental settings for encoder pre-training

The encoders of the baseline methods are pre-trained on the large ImageNet-1K Russakovsky et al. (2015) dataset and its subset ImageNet-100 Tian et al. (2020a), under two different backbone encoder architectures ResNet-50 and ResNet-34 He et al. (2016). The encoders of MoCo and SimSiam based methods are pre-trained for 200 epochs, while SupCon's encoder undergoes 100 epochs of pre-training. Batch size of all pre-training experiments is set to 256. We follow Khosla et al. (2020); Chen et al. (2020b); Chen & He (2021) for other settings of these baseline methods.

For the sake of fair comparison, pre-training with `CoCor` incorporated follows the same experiment settings as their corresponding baselines. The MMNN in `CoCor` is a 3-layer MLP with monotonic linear embedding layers You et al. (2017) and ReLU units. While most of the results are demonstrated when using only 1% of the pre-training dataset labels for tuning the MMNN, we also demonstrate the good performance of `CoCor` in an ablation study where the labeled data usage is significantly further reduced in Table 5.4. `CoCor` utilizes a basic data augmentation set $\Omega_a$, consisting of 14 commonly used augmentation functions as detailed in Appendices. We impose the DA consistency constraint to several composite DA sets $\Omega_c^{<l>}$ of specific lengths. For most pre-training experiments, we take $l = 1, 2, 3$, i.e., for each example, we sample three composite DAs of length 1, 2, and 3, and apply them to get three views of data to calculate the consistency loss. More details of data augmentations and pre-training setup are provided in Appendices.

### 5.2 Main Results

**Linear evaluation** Each pre-trained encoder is parameter-frozen and paired with a linear classifier, which is fine-tuned, following the linear evaluation protocol of Krizhevsky et al. (2017), as detailed in Appendices. Linear evaluation is conducted on the following datasets: Cifar-10/100 Krizhevsky et al. (2009), CUB-

Table 1: Top-1 accuracies (%) of linear evaluation. All ResNet-50 backbone encoders are pre-trained on ImageNet-100.

| Method | Epochs | IN-100 | Cifar10 | Cifar100 | CUB200 | Caltech101 | SUN397 | Food101 | Flowers102 | Pets |
|---|---|---|---|---|---|---|---|---|---|---|
| MoCo Chen et al. (2020b) | 200 | 67.04 | 82.15 | 59.17 | 21.52 | 79.23 | 39.43 | 52.97 | 71.77 | 56.94 |
| MoCo + CoCor (**Ours**) | 200 | **71.66** | **83.87** | **59.64** | **22.57** | **81.51** | **41.95** | **54.81** | **75.65** | **60.23** |
| SimSiam Chen & He (2021) | 200 | 73.92 | 84.66 | 61.82 | 26.13 | 85.11 | 45.96 | 58.73 | 82.40 | 65.60 |
| SimSiam + CoCor (**Ours**) | 200 | **83.70** | **86.89** | **66.36** | **34.50** | **87.85** | **49.92** | **62.48** | **87.95** | **76.04** |
| SupCon Khosla et al. (2020) | 100 | 80.16 | 84.30 | 61.33 | 26.46 | 85.04 | 41.95 | 52.45 | 76.99 | 73.75 |
| SupCon + CoCor (**Ours**) | 100 | **82.14** | **85.49** | **62.18** | **27.40** | **87.12** | **42.87** | **53.93** | **79.04** | **75.95** |

Table 2: Top-1 accuracies (%) of linear evaluation. All ResNet-50 backbone encoders are pre-trained on ImageNet-1K.

| Method | Epochs | IN-1K | Cifar10 | Cifar100 | CUB200 | Caltech101 | SUN397 | Food101 | Flowers102 | Pets |
|---|---|---|---|---|---|---|---|---|---|---|
| MoCo Chen et al. (2020b) | 200 | 67.56 | 92.24 | 74.33 | 41.54 | 92.00 | 58.28 | 69.84 | 88.86 | 81.74 |
| MoCo + CoCor (**Ours**) | 200 | **72.83** | **93.28** | **76.19** | **44.11** | **93.10** | **60.15** | **71.10** | **90.39** | **83.08** |
| SimSiam Chen & He (2021) | 200 | 70.93 | 93.08 | 76.24 | 51.05 | 93.13 | 60.81 | 71.18 | 92.11 | 85.01 |
| SimSiam + CoCor (**Ours**) | 200 | **73.25** | **94.04** | **78.12** | **54.50** | **94.51** | **63.09** | **73.31** | **92.84** | **87.24** |
| SupCon Khosla et al. (2020) | 100 | 74.18 | 93.17 | 76.55 | 52.30 | 93.28 | 61.03 | 71.54 | 91.34 | 85.35 |
| SupCon + CoCor (**Ours**) | 100 | **76.24** | **94.61** | **77.91** | **55.73** | **94.66** | **63.80** | **74.09** | **92.17** | **87.21** |

200 Wah et al. (2011), Caltech-101 Fei-Fei et al. (2004), SUN397 Xiao et al. (2010), Food101 Bossard et al. (2014), Flowers102 Nilsback & Zisserman (2008), Oxford-IIIT Pet (Pets) Parkhi et al. (2012), Aircraft Maji et al. (2013), and StanfordCars Krause et al. (2013).

The classification accuracies for ImageNet-100 and ImageNet-1K pre-trained encoders are presented in Table 1 and Table 2. Notably, `CoCor` demonstrates a substantial enhancement in the performance of the pre-trained encoder across diverse classification tasks. This improvement suggests that `CoCor` contributes to an enhanced generalizability of the encoder, achieved by incorporating a wider range of augmentations and by complementing the semantics within the feature space.

**Object detection**  We fine-tune the pre-trained encoders on VOC2007+2012 Everingham et al. (2009) and COCO2017 Lin et al. (2014) datasets for object detection downstream tasks. The pre-trained encoders are converted to generalized R-CNN Girshick et al. (2014) detectors with a ResNet50-C4 backbone by using Detectron2 Wu et al. (2019). The models trained on VOC2007 and COCO2007 are subsequently evaluated on the corresponding dataset's test set.

In comparison with the baselines, Table 3 shows that `CoCor` substantially improves accuracy under standard metrics such as $AP$, $AP_{50}$, and $AP_{75}$, suggesting that `CoCor` is effective in achieving the pre-trained encoders' transferability. Notably, `CoCor` shows substantial improvements in detecting challenging small and medium object detection, as evidenced by the increased $AP_s$ and $AP_m$ scores.

**Post-hoc study**  We visualize the effect of the proposed consistency loss of equation 2 during pre-training by comparing the latent similarity of encoders pre-trained with and without the consistency loss in Table 4 (a), showing the averaged latent similarity of each specific length of composite augmentations on the ImageNet-100 dataset. Clearly, the use of `CoCor` results in larger latent similarity values across a wide range of DA lengths, demonstrating a tighter clustering of image views with the same identify. This aligns with the analysis presented in Wang & Isola (2020), which, even without taking into account the dependency on augmentation strength, suggests that a stronger clustering of views generated by weaker augmentations is correlated with improved encoder performance. Additionally, the image retrieval task results in Figure 1(b) suggest that `CoCor` learns a more semantically consistent feature space compared to the baseline methods.

Table 3: Transfer learning results on VOC and COCO object detection tasks.

| Method | Pre-train Epochs | VOC07+12 AP | $AP_{50}$ | $AP_{75}$ | COCO $AP$ | $AP_{50}$ | $AP_{75}$ | $AP_s$ | $AP_m$ | $AP_l$ |
|---|---|---|---|---|---|---|---|---|---|---|
| MoCo He et al. (2020) | 200 | 50.23 | 76.52 | 54.42 | 34.23 | 55.08 | 36.60 | 16.67 | 37.27 | 50.97 |
| MoCo + CoCor (**Ours**) | 200 | **51.11** | **77.41** | **54.88** | **39.13** | **58.30** | **42.19** | **23.19** | **43.45** | **52.13** |
| SimSiam Chen & He (2021) | 200 | 52.04 | 77.86 | 57.11 | 34.15 | 55.21 | 36.50 | 15.30 | 37.37 | 50.49 |
| SimSiam + CoCor (**Ours**) | 200 | **54.31** | **80.60** | **59.58** | **34.80** | **56.29** | **37.05** | **17.53** | **39.88** | **51.74** |
| SupCon Khosla et al. (2020) | 100 | 47.41 | 75.83 | 53.17 | 34.26 | 55.11 | 36.35 | 14.34 | 37.40 | 50.56 |
| SupCon + CoCor (**Ours**) | 100 | **49.03** | **77.22** | **54.19** | **35.67** | **56.10** | **37.98** | **16.63** | **38.71** | **51.29** |

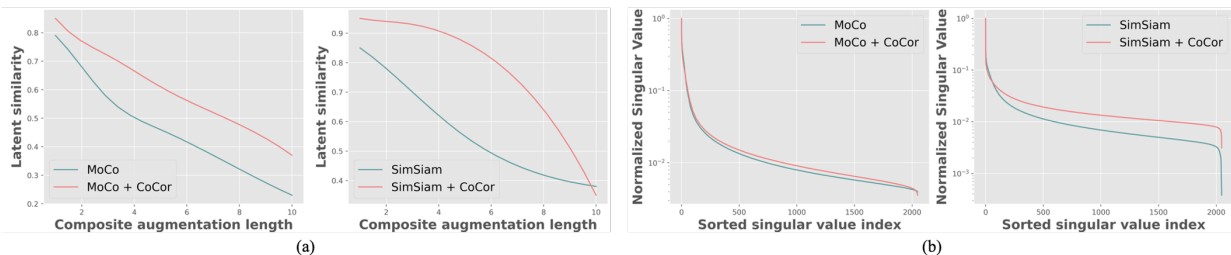

(a)          (b)

Figure 4: (a): Latent similarities under different composite augmentation lengths of encoders pre-trained with and without consistency loss, and (b) singular values of the learned latent presentations, both evaluated on ImageNet-100.

This introduced consistency ensures that data sharing similar latent semantics is positioned close, while dissimilar instances are kept distant in the feature space.

**Dimensional collapse evaluation** We illustrate the singular values of the latent representations of pre-trained encoders on ImageNet-100 by conducting principal component analysis (PCA) on the feature vectors. The occurrence of dimensional collapse is typically associated with observed small singular values. However, as depicted in Figure 4 (b), this is less likely to be the case for the encoders pre-trained by `CoCor` compared with the two baselines. While stronger data augmentations Jing et al. (2021) and larger datasets Li et al. (2022a) have been suggested to contribute to dimensional collapse, this issue is not observed in the `CoCor` encoders. In conventional contrastive learning methods, the encoder is forced to be invariant to feature variance introduced by augmentations, potentially leading to constant values in some feature space dimensions. The dimensional collapse problem is mitigated in `CoCor`, possibly due to the imposed consistency that captures this variance.

Table 4: Comparison of top-1 accuracies (%) in linear evaluation of SimSiam, existing augmentation-aware methods, and `CoCor`. Encoders (ResNet-34) are trained on ImageNet-100 for 200 epochs.

| Method | ImageNet100 | CUB200 | Flowers102 | StanfordCars |
|---|---|---|---|---|
| SimSiam Chen & He (2021) | 63.90 | 29.10 | 76.10 | 19.60 |
| Augself Lee et al. (2021) | 75.96 | 29.96 | 82.96 | 29.41 |
| Hierarchical Zhang & Ma (2022) | 67.10 | 33.90 | 83.60 | 20.70 |
| EquiMod Devillers & Lefort (2023) | 65.45 | 30.09 | 77.91 | 19.06 |
| SimSiam+CoCor (**Ours**) | **77.76** | **34.07** | **83.88** | **29.93** |

Table 5: Comparison of top-1 accuracies (%) in linear evaluation of methods that leverage stronger data augmentations. Encoders (ResNet-50) are trained on ImageNet-1K for 200 epochs.

| Method | ImageNet1K | CUB200 | Flowers102 | Aircraft |
|---|---|---|---|---|
| CLSA Wang & Qi (2021) | 69.4 | 40.20 | 89.97 | 61.24 |
| RényiCL Lee & Shin (2023) | 72.6 | **45.05** | 90.17 | 61.80 |
| MoCo+CoCor (**Ours**) | **72.8** | 44.11 | **90.39** | **63.18** |

## 5.3 Comparison with the state-of-the-art

**Comparison with augmentation-aware methods**   Recently, several works Devillers & Lefort (2023); Lee et al. (2021); Zhang & Ma (2022) have taken into account augmentation-aware information in contrastive learning. AugSelf Lee et al. (2021) achieves this by training an auxiliary network to predict the difference between augmentations applied to generate positive pairs. Zhang & Ma (2022) encodes an augmented image along with the DA parameters using two networks, and combines the two embeddings to form a feature vector for contrastive loss. EquiMod Devillers & Lefort (2023) trains a network to predict the representation of an augmented view from the original data and the applied DA. A performance comparison between these methods and `CoCor` on linear evaluation is presented in Table 4. All the encoders have a ResNet-34 architecture and are pre-trained on ImageNet-100 for 200 epochs. Compared to Lee et al. (2021); Zhang & Ma (2022); Devillers & Lefort (2023), `CoCor` introduces much stronger DA and more diverse DA types. `CoCor` consistently outperforms them, which may be attributed to its effective utilization of a significantly broader set of augmentations while maintaining the well-specified monotonic property in augmentation strength.

**Comparison with methods using stronger data augmentations**   Several recent contrastive learning approaches incorporate the use of stronger data augmentations. CLSA Wang & Qi (2021) employs the distribution divergence between weakly augmented views as supervision for the divergence between a weakly and a strongly augmented view. RényiCL Lee & Shin (2023) proposes a new contrastive objective using Rényi divergence to address strongly augmented data. Table 5 shows the comparison between these methods and ours. Here all encoders are ResNet-50 trained for 200 epochs on ImageNet-1K. Unlike RényiCL and CLSA, which do not explicitly distinguish between different types of data augmentations (DAs), `CoCor` addresses this by using the MMNN to differentiate various DA types. Furthermore, `CoCor` introduces DAs that are not only stronger but also more diverse, implementing consistency across different lengths of DAs, in contrast to RényiCL and CLSA, which apply a single length of DA per pre-training process.

## 5.4 Ablation Studies

**Basic Data Augmentations**   define the augmentation-aware information to be learned by the encoder. To better understand the role of these data augmentations during pre-training, we separate them into two groups: $\Omega_{color}$ and $\Omega_{affine}$. $\Omega_{color}$ includes augmentations that change the color of each pixel while maintaining its position, such as `Brightness` and `Contrast`. $\Omega_{affine}$ comprises affine transformations, such as `Rotate` and `Shear`. Composition of $\Omega_{color}$ and $\Omega_{affine}$ is provided in the Appendices. Using `CoCor`, we pre-trained three encoders for 500 epochs on ImageNet-100 with $\Omega_{color}$, $\Omega_{affine}$, and $\Omega_{color} \bigcup \Omega_{affine}$, respectively. Table 6 presents the linear evaluation results on ImageNet-100 and two fine-grained datasets, CUB-200 and Flowers102. The results indicate that `CoCor` achieves better overall performance than LooC Xiao et al. (2020) and AugSelf Lee et al. (2021), which have improved transferability to various downstream recognition tasks. Notably, while $\Omega_{color} \bigcup \Omega_{affine}$ delivers the best overall results across the tested datasets, refining the basic data augmentations can enhance `CoCor`'s performance on specific downstream tasks. For example, color-aware information is crucial for discriminating flower images in the Flower102 dataset Nilsback & Zisserman (2008), whereas position-related information introduced by affine transformations is less relevant. Thus, applying only color-related augmentations while maintaining invariance to affine transformations improves performance on Flower102.

Table 6: Comparison between LooC, AugSelf, and `CoCor`. `CoCor` is pre-trained using basic data augmentation pools, which include color-related augmentations, affine transformations, and their combination. Top-1 Linear evaluation accuracy is reported. Encoders (ResNet-50) are trained on ImageNet-100 for 500 epochs.

| Method | ImageNet100 | CUB200 | Flowers(5-shot) | Flowers(10-shot) |
|---|---|---|---|---|
| MoCo* Chen et al. (2020b) | 81.0 | 36.7 | 67.9 | 77.3 |
| LooC(color)* Xiao et al. (2020) | 81.1 | 40.1 | 68.2 | 77.6 |
| LooC(rotation)* Xiao et al. (2020) | 80.2 | 38.8 | 70.1 | 79.3 |
| LooC(color, rotation)* Xiao et al. (2020) | 79.2 | 39.6 | 70.9 | 80.8 |
| MoCo + AugSelf* Lee et al. (2021) | 82.4 | 37.0 | 81.7 | 84.5 |
| SimSiam + AugSelf* Lee et al. (2021) | 82.6 | 45.3 | 86.4 | 88.3 |
| SimSiam + CoCor (color) | 83.5 | 46.5 | **89.1** | **90.2** |
| SimSiam + CoCor (affine) | 81.8 | 45.2 | 83.5 | 86.1 |
| SimSiam + CoCor (color+affine) | **83.7** | **47.9** | 88.9 | 89.6 |

*Since the official implementation of LooC has not been released, we adopt the results from Lee et al. (2021); Xiao et al. (2020). To ensure a fair comparison, we maintain the same experimental settings for all methods in this table.

Table 7: Linear evaluation of CoCor models trained with and without MMNN. Encoders (ResNet-50) are trained on ImageNet-100 for 200 epochs.

| | baseline | w/o MMNN | w/ MMNN |
|---|---|---|---|
| MoCo + CoCor | 67.04 | 69.51 | **71.66** |
| SimSiam + CoCor | 73.92 | 82.18 | **83.70** |

**Effectiveness of the MMNN**  `CoCor` utilizes an MMNN model to learn the optimal latent similarities $l_d^*$ of various composite augmentations. To see the benefits of this learnable MMNN model, we replace it by a manually optimized model: we manually search for the best $l_d^*$ value for each composite augmentation length, determined by extensive trial-and-error, to achieve optimal linear evaluation performance. More detailed experimental settings and results on $l_d^*$ selection are included in Appendices.

We pre-train ResNet-50 encoders built upon SimSiam and MoCo with the MMNN and the manual model based on composite augmentation lengths $l = 1, 2, 3$, on ImageNet-100 for 200 epochs. The resulting performances are reported in Table 7, which also includes the performances of the MoCo and SimSiam baselines. There is the clear advantages of the encoders trained with the MMNN over the ones without, i.e. with the manual model.

**Effect of the amount of labeled data in MMNN training**  We further study the impact of the amount of supervision used to train the MMNN on the encoder performance. We pre-train ResNet-50 `CoCor` encoders built upon SimSiam on ImageNet-100 for 100 epochs while training the MMNN using 100%, 10%, 1%, and 0.1% of the labeled data in the dataset. The pre-trained backbones are evaluated by linear evaluation on ImageNet-100. Table 8 shows that drastically reducing the amount of labeled MMNN training data from 100% to 0.1% only results in a 0.94% performance drop, and all four `CoCor` models outperform the baseline SimSiam by more than 15%.

**Composite augmentation length in pre-training**  The composite augmentations introduced in our work produce diverse informative views. We test the effect of the strength of composite augmentations by running experiments where only one length of composite augmentation is adopted per pre-training experiment. We run `CoCor` on SimSiam and MoCo with single augmentation length 1, 2, 3, 4, and a combination of lengths 1, 2, and 3. All encoders are ResNet-50 pre-trained on ImageNet-100 for 50 epochs, and are then evaluated on ImageNet-100.

As seen in Table 5, `CoCor` is able to leverage augmentations of varying lengths, and combining DAs at all these lengths further improves performance. Different from the results presented in  Chen et al. (2020a);

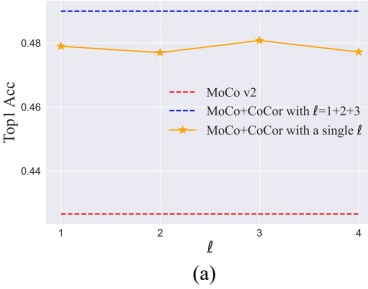 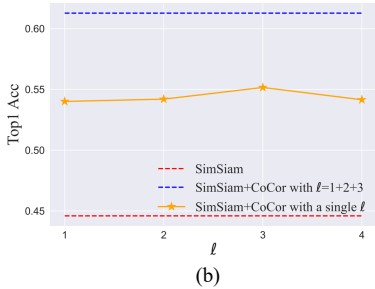

(a)  (b)

Figure 5: Linear evaluation of encoders trained by CoCor on (a) MoCo and (b) SimSiam with single length *l* of composite augmentations and a combination of length 1, 2, and 3.

Table 8: Performance of CoCor models trained with different amounts of labeled data. Encoders (ResNet-50) are trained on ImageNet-100 for 50 epochs.

| Percentage of labels | 100% | 10% | 1% | 0.1% | SimSiam |
|---|---|---|---|---|---|
| Top-1 Acc | 75.97 | 75.82 | 75.79 | 75.46 | 59.20 |

Tian et al. (2020b), which show that strong augmentations can degrade performance, we have conducted additional experiments with much stronger composite augmentations to demonstrate `CoCor`'s ability in leveraging such augmentations. These results, presented in the Appendices, demonstrate `CoCor`'s capability to benefit from even stronger augmentations compared to recent works that introduce strong augmentations in their methodsLee & Shin (2023); Wang & Qi (2021).

## 6 Discussion

In this paper, we introduce `CoCor`, a systematic approach to exploring diverse data augmentations in contrastive learning. Our contribution includes the introduction of DA consistency to quantify the dependency of the desired latent similarity on the applied data augmentation. We propose a consistency loss to guide the encoder training towards improved DA consistency. To enforce the DA consistency, we employ a data-driven method to learn the optimal dependency and apply it to the encoder training. The effectiveness of `CoCor` is substantiated through extensive experimental results on various tasks and datasets. Further details on the MMNN update rule, additional experimental settings, and extended results are provided in the Appendices.

As a potential future direction, we suggest exploring the learning of optimal latent similarity with respect to data examples. Specifically, extending the MMNN to take data instances as input could allow for considering the variance of latent similarity caused by data variance. This approach has the potential to learn a more accurate mapping from data augmentation to latent similarity.

## 7 Acknowledgement

This material is based upon work supported by the National Science Foundation under Grant No. 1956313.

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

## A Improving CoCor with an Alternative Consistency Loss

The original consistency loss is defined as:

$$\mathcal{L}_{\text{consistent}}(\boldsymbol{\theta_e}|\boldsymbol{\theta_d}) = \mathbb{E}_{\boldsymbol{x}\sim\mathcal{X}, a_c\sim\boldsymbol{\Omega_c}}\left[|l_d(\boldsymbol{x}; f_{\boldsymbol{\theta_e}}, a_c) - g_{\boldsymbol{\theta_d}}(\boldsymbol{v_c}(a_c))|\right] \tag{6}$$

However, in our preliminary experiments, we find the following consistency loss leads to better downstream tasks performance than equation 6.

$$\mathcal{L}_{\text{consistent}}(\boldsymbol{\theta_e}|\boldsymbol{\theta_d}) = \mathbb{E}_{\boldsymbol{\Omega_c^{<l>}}\sim\boldsymbol{\Omega_c}}[\text{softplus}[\mathbb{E}_{\boldsymbol{x}\sim\mathcal{X}, a_{c,i}^{<l>}\sim\boldsymbol{\Omega_c^{<l>}}}[l_d(\boldsymbol{x}; f, a_{c,i}^{<l>}) - g_{\boldsymbol{\theta_d}}(\boldsymbol{v_c}(a_{c,i}^{<l>}))]]] \tag{7}$$

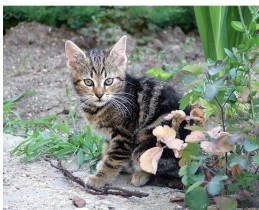 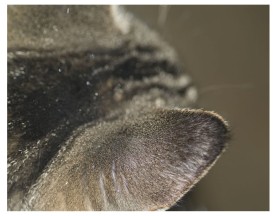 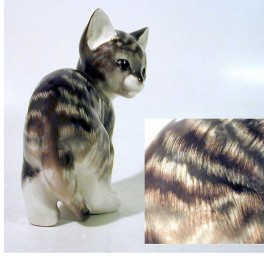 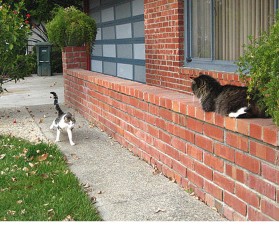

Figure 6: Images from a same category of a dataset. Target items may be of different sizes and colors, and may be located at different places in images. Thus same augmentation can have different effects on different images. ImageNet is used for illustration.

The motivation is to reduce the high variance among images in visual datasets. The exactly same composite augmentation hence can lead to different latent similarity in the feature space. For instance, as illustrated in Figure 6, target items of images may be of different size and shape, and may be located at different places. Therefore, enforcing all augmented views to share the same optimal similarity may not lead to the best performance. To this end, equation 7 calculates the averaged latent similarity in a minibatch of each length of augmentation composition. Therefore, equation 7 imposes DA consistency constraint on the averaged latent similarity in a batch instead of on each image, which help better handle the variance.

Additionally, data augmentation that can greatly distort an images may not be able to largely distort the identity of another one. However, the absolute value loss pushes apart those less distorted views from the representations of the original data. Thus, softplus is used to replace the absolute value calculation because softplus produce small gradient for negative numbers.

## B Derivation of the Update Rule of the MMNN

In our implementation, the parameter of the encoder ($\boldsymbol{\theta_e}$) and the parameter of MMNN ($\boldsymbol{\theta_d}$) is updated in an alternate manner, as in equation 8.

$$\begin{aligned}
\boldsymbol{\theta_e'} &= \boldsymbol{\theta_e} - \eta_e \cdot \nabla_{\boldsymbol{\theta_e}}\mathcal{L}_u(\boldsymbol{\theta_e}|\boldsymbol{\theta_d}) \\
\boldsymbol{\theta_d'} &= \boldsymbol{\theta_d} - \eta_d \cdot \nabla_{\boldsymbol{\theta_d}}\text{CE}(\boldsymbol{\theta_e'}(\boldsymbol{\theta_d}))
\end{aligned} \tag{8}$$

With the overall loss function to optimize the encoder, the update of $\boldsymbol{\theta_e}$ with learning rate $\eta_e$ can be written as:

$$\boldsymbol{\theta_e'} = \boldsymbol{\theta_e} - \eta_e \cdot (\nabla_{\boldsymbol{\theta_e}}\mathcal{L}_{\text{contrast}} + \nabla_{\boldsymbol{\theta_e}}\mathcal{L}_{\text{consistent}}) \tag{9}$$

By applying chain rule, the update of $\boldsymbol{\theta_d}$ with learning rate $\eta_d$ can be rewritten as:

$$\boldsymbol{\theta_d'} = \boldsymbol{\theta_d} - \eta_d \cdot (\nabla_{\boldsymbol{\theta_d}} \text{CE}(\boldsymbol{\theta_e'}(\boldsymbol{\theta_d})),$$

$$\text{where}: \quad \nabla_{\boldsymbol{\theta_d}} \text{CE}(\boldsymbol{\theta_e'}(\boldsymbol{\theta_d})) = \frac{\partial}{\partial \boldsymbol{\theta_e'}} \text{CE}(\boldsymbol{\theta_e'}) \cdot \frac{\partial \boldsymbol{\theta_e'}}{\partial \boldsymbol{\theta_d}} \tag{10}$$

In equation 10, $\frac{\partial}{\partial \boldsymbol{\theta_e'}} \text{CE}(\boldsymbol{\theta_e'})$ can be computed via back-propagation. Thus, we focus on the computation of $\frac{\partial \boldsymbol{\theta_e'}}{\partial \boldsymbol{\theta_d}}$.

First, we substitute $\boldsymbol{\theta_e'}$ in equation 10 with the one-step updated $\boldsymbol{\theta_e'}$ in equation 9, following the approach used in Pham et al. (2021); Yin et al. (2023). We also neglect the higher order dependency of $\boldsymbol{\theta_e}$ on $\boldsymbol{\theta_d}$ as adopted in Pham et al. (2021). $\frac{\partial \boldsymbol{\theta_e'}}{\partial \boldsymbol{\theta_d}}$ can thus be expanded as below:

$$\begin{aligned}
\frac{\partial \boldsymbol{\theta_e'}}{\partial \boldsymbol{\theta_d}} &= \frac{\partial}{\partial \boldsymbol{\theta_d}} (\boldsymbol{\theta_e} - \eta_e \cdot (\nabla_{\boldsymbol{\theta_e}} \mathcal{L}_{\text{contrast}} + \nabla_{\boldsymbol{\theta_e}} \mathcal{L}_{\text{consistent}})) \\
&= -\eta_e \cdot \frac{\partial}{\partial \boldsymbol{\theta_d}} \frac{\partial}{\partial \boldsymbol{\theta_e}} \mathcal{L}_{\text{consistent}} \\
&= -\eta_e \cdot \frac{\partial}{\partial \boldsymbol{\theta_d}} \frac{\partial}{\partial \boldsymbol{\theta_e}} \mathbb{E}_{l \in \ell} [\text{softplus}[\mathbb{E}_{i \in \mathcal{B}} \ [(f_{\boldsymbol{\theta_e}}(\boldsymbol{x_i^l})^T \cdot f_{\boldsymbol{\theta_e}}(\boldsymbol{x_i})) - g_{\boldsymbol{\theta_d}}(\boldsymbol{v_c}(a_{c,i}^{<l>}))]]]
\end{aligned} \tag{11}$$

where $\boldsymbol{x_i^l} = a_{c,i}^{<l>}(\boldsymbol{x_i})$ denotes the augmented data, $\ell$ denotes the set of lengths of composite augmentations that are used, and $\mathcal{B}$ denotes the set of data indices in a minibatch.

For simplification in later section, we define:

$$\begin{aligned}
k^l &= \mathbb{E}_{i \in \mathcal{B}}[(f_{\theta_e}(\boldsymbol{x_i^l})^T \cdot f_{\boldsymbol{\theta_e}}(\boldsymbol{x_i})) - g_{\boldsymbol{\theta_d}}(\boldsymbol{v_c}(a_{c,i}^{<l>}))] \\
\text{sim}^l &= \mathbb{E}_{i \in \mathcal{B}}[(f_{\boldsymbol{\theta_e}}(\boldsymbol{x_i^l})^T \cdot f_{\boldsymbol{\theta_e}}(\boldsymbol{x_i}))]
\end{aligned} \tag{12}$$

We then combine equation 11 with equation 12:

$$\begin{aligned}
\frac{\partial \boldsymbol{\theta_e'}}{\partial \boldsymbol{\theta_d}} &= -\eta_e \cdot \mathbb{E}_{l \in \ell} [\frac{\partial}{\partial \boldsymbol{\theta_d}} (\frac{-e^{k^l}}{1 + e^{k^l}} \cdot \frac{\partial}{\partial \boldsymbol{\theta_e}} \text{sim}^l)] \\
&= -\eta_e \cdot \mathbb{E}_{l \in \ell} [\frac{e^{k^l}}{(1 + e^{k^l})^2} \cdot \frac{\partial}{\partial \boldsymbol{\theta_e}} \text{sim}^l \cdot \frac{\partial}{\partial \boldsymbol{\theta_d}} \mathbb{E}_{i \in \mathcal{B}}[g_{\boldsymbol{\theta_d}}(\boldsymbol{v_c}(a_{c,i}^{<l>}))]]]
\end{aligned} \tag{13}$$

By incorporating equation 13 with equation 10, we have:

$$\nabla_{\boldsymbol{\theta_d}} \text{CE}(\boldsymbol{\theta_e'}(\boldsymbol{\theta_d})) = \frac{\partial}{\partial \boldsymbol{\theta_e'}} \text{CE}(\boldsymbol{\theta_e'}) \cdot \ (-\eta_e \cdot \mathbb{E}_{l \in \ell} [\frac{e^{k^l}}{(1 + e^{k^l})^2} \cdot \frac{\partial}{\partial \boldsymbol{\theta_e}} \text{sim}^l \cdot \frac{\partial}{\partial \boldsymbol{\theta_d}} \mathbb{E}_{i \in \mathcal{B}}[g_{\boldsymbol{\theta_d}}(\boldsymbol{v_c}(a_{c,i}^{<l>}))]]]) \tag{14}$$

Additionally, in equation 14, $\frac{\partial}{\partial \boldsymbol{\theta_e'}} \text{CE}(\boldsymbol{\theta_e'})$ can be approximated with first order Taylor expansion:

$$\text{CE}(\boldsymbol{\theta_e'}) - \text{CE}(\boldsymbol{\theta_e}) \approx (\boldsymbol{\theta_e'} - \boldsymbol{\theta_e}) \cdot \frac{\partial}{\partial \boldsymbol{\theta_e'}} \text{CE}(\boldsymbol{\theta_e'}) \tag{15}$$

Thus, we have:

$$\frac{\partial}{\partial \boldsymbol{\theta_e'}} \text{CE}(\boldsymbol{\theta_e'}) \approx \frac{\text{CE}(\boldsymbol{\theta_e'}) - \text{CE}(\boldsymbol{\theta_e})}{\boldsymbol{\theta_e'} - \boldsymbol{\theta_e}} = \frac{\text{CE}(\boldsymbol{\theta_e'}) - \text{CE}(\boldsymbol{\theta_e})}{-\eta_e \cdot (\nabla_{\boldsymbol{\theta_e}} \mathcal{L}_{\text{contrast}} + \nabla_{\boldsymbol{\theta_e}} \mathcal{L}_{\text{consistent}})} \tag{16}$$

We can now rewrite equation 14 as:

$$\nabla_{\boldsymbol{\theta_d}}\text{CE}(\boldsymbol{\theta'_e}(\boldsymbol{\theta_d})) \approx \frac{\text{CE}(\boldsymbol{\theta'_e}) - \text{CE}(\boldsymbol{\theta_e})}{(\nabla_{\boldsymbol{\theta_e}}\mathcal{L}_{\text{contrast}} + \nabla_{\boldsymbol{\theta_e}}\mathcal{L}_{\text{consistent}})} \cdot \mathbb{E}_{l \in \ell}\big[\frac{e^{k^l}}{(1 + e^{k^l})^2} \cdot \frac{\partial}{\partial \boldsymbol{\theta_e}}\text{sim}^l \cdot \frac{\partial}{\partial \boldsymbol{\theta_d}}\mathbb{E}_{i \in \mathcal{B}}[g_{\boldsymbol{\theta_d}}(\boldsymbol{v_c}(a_{c,i}^{<l>}))]\big] \tag{17}$$

Further more, equation 17 can be further simplified. Note that $\nabla_{\boldsymbol{\theta_e}}\mathcal{L}_{\text{contrast}} + \nabla_{\boldsymbol{\theta_e}}\mathcal{L}_{\text{consistent}}$ can be approximated by:

$$\nabla_{\boldsymbol{\theta_e}}\mathcal{L}_{\text{contrast}} + \nabla_{\boldsymbol{\theta_e}}\mathcal{L}_{\text{consistent}} := \nabla_{\boldsymbol{\theta_e}}\mathcal{L}_u(\boldsymbol{\theta_e}) \approx \frac{\mathcal{L}_u(\boldsymbol{\theta'_e}) - \mathcal{L}_u(\boldsymbol{\theta_e})}{\boldsymbol{\theta'_e} - \boldsymbol{\theta_e}} \tag{18}$$

Similarly, the approximation of $\frac{\partial}{\partial \boldsymbol{\theta_e}}\text{sim}^l$ can be written as:

$$\frac{\partial}{\partial \boldsymbol{\theta_e}}\text{sim}^l \approx \frac{\text{sim}^l(\boldsymbol{\theta'_e}) - \text{sim}^l(\boldsymbol{\theta_e})}{\boldsymbol{\theta'_e} - \boldsymbol{\theta_e}}$$
$$\text{where:} \quad \text{sim}^l(\boldsymbol{\theta'_e}) = \mathbb{E}_{i \in \mathcal{B}}[(f_{\boldsymbol{\theta'_e}}(\boldsymbol{x_i^l})^T \cdot f_{\boldsymbol{\theta'_e}}(\boldsymbol{x_i}))] \tag{19}$$

Finally, by combining equation 18 and equation 19 with equation 17, $\nabla_{\boldsymbol{\theta_d}}\text{CE}(\boldsymbol{\theta'_e}(\boldsymbol{\theta_d}))$ can be rewritten as follows:

$$\nabla_{\boldsymbol{\theta_d}}\text{CE}(\boldsymbol{\theta'_e}(\boldsymbol{\theta_d})) \approx (\text{CE}(\boldsymbol{\theta'_e}) - \text{CE}(\boldsymbol{\theta_e})) \cdot \mathbb{E}_{l \in \ell}\big[\frac{e^{k^l}}{(1 + e^{k^l})^2} \cdot \frac{\text{sim}^l(\boldsymbol{\theta'_e}) - \text{sim}^l(\boldsymbol{\theta_e})}{\mathcal{L}_u(\boldsymbol{\theta'_e}) - \mathcal{L}_u(\boldsymbol{\theta_e})} \cdot \frac{\partial}{\partial \boldsymbol{\theta_d}}\mathbb{E}_{i \in \mathcal{B}}[g_{\theta_d}(\boldsymbol{V}(A))]\big] \tag{20}$$

Note that in equation 20, all the derivative terms can be calculated directly via back propagation, resulting in a scalable algorithm.

## C    Candidate Augmentations in CoCor

The set of basic augmentations $\Omega_a$ used to form composite augmentation consists of 14 different types data augmentations, i.e., $\Omega_a = \{$AutoContrast, Brightness, Color, Contrast, Rotate, Equalize, Identity, Posterize, Sharpness, ShearX, ShearY, Solarize, TranslateX, TranslateY$\}$.

In Section 5.4, we conduct ablation studies on the composition of basic data augmentations, as illustrated in Table 6. In these experiments, $\Omega_a$ is divided into two sets: $\Omega_{\text{color}}$ and $\Omega_{\text{affine}}$. The compositions of the two sets are as follows: $\Omega_{\text{color}} = \{$AutoContrast, Brightness, Color, Contrast, Equalize, Identity, Posterize, Sharpness, Solarize,$\}$ and $\Omega_{\text{affine}} = \{$Rotate, Identity, ShearX, ShearY, TranslateX, TranslateY$\}$.

## D    Experimental Setup for Pre-training

All pre-training experiment are conducted using stochastic gradient descent (SGD) as the optimizer. A cosine decay scheduler is adopted for scheduling the learning rate. All pre-training adopt a training data batch size of 256. Detailed setups used for each method are as follows.

- **MoCo** He et al. (2020); Chen et al. (2020b): We use a starting learning rate of $3 \times 10^{-2}$, a weight decay of $1 \times 10^{-4}$, and a momentum of 0.9 for the optimizer. For the dual encoders of MoCo, feature space dimension is 128, memory queue size is 65536, and the momentum of updating the key encoder is 0.999. For the consistency loss, the query encoder takes both the original data and its augmented view as input. However, the query encoder is stop-gradient when encoding the original input. The encoder receives gradient while encoding the augmented views.

- **SimSiam** Chen & He (2021): A starting learning rate of $5 \times 10^{-2}$, a weight decay of $1 \times 10^{-4}$, and a momentum of 0.9 are used for the SGD optimizer. Predictor and projector and output dimension are both 2048. For the consistency loss, representations of the original data are obtained by letting the data go through the stop-gradient backbone and the projector. The augmented views go through backbone, projector, and predictor for the representations of them. All the three models receive gradient while calculating the representations of augmented views.

- **SupCon** Khosla et al. (2020): We use a starting learning rate of $3 \times 10^{-2}$, a weight decay of $1 \times 10^{-4}$, and a momentum of 0.9 for the optimizer. Feature space dimension of the projection head output is 128. In the consistency loss, the encoder takes both the original data and its augmented view as input. The encoder is stop-gradient when encoding the original input. The encoder receives gradient while calculating representations of the augmented views.

## E  Experimental Setup for Linear evaluation

We follow the linear evaluation protocol of Chen et al. (2020a); Lee et al. (2021); Kornblith et al. (2019). A one-layer linear classifier is trained upon the frozen pre-trained backbone. `Resize`, `CentorCrop`, and `Normalization` are used as the training transformations. An L-BFGS optimizer is adopted for minimizing the $\ell_2$-regularized cross-entropy loss. The optimal regularization parameter is selected on the validation set of each dataset. Finally, models are trained with the optimal regularization parameter, and the obtained test accuracies are reported.

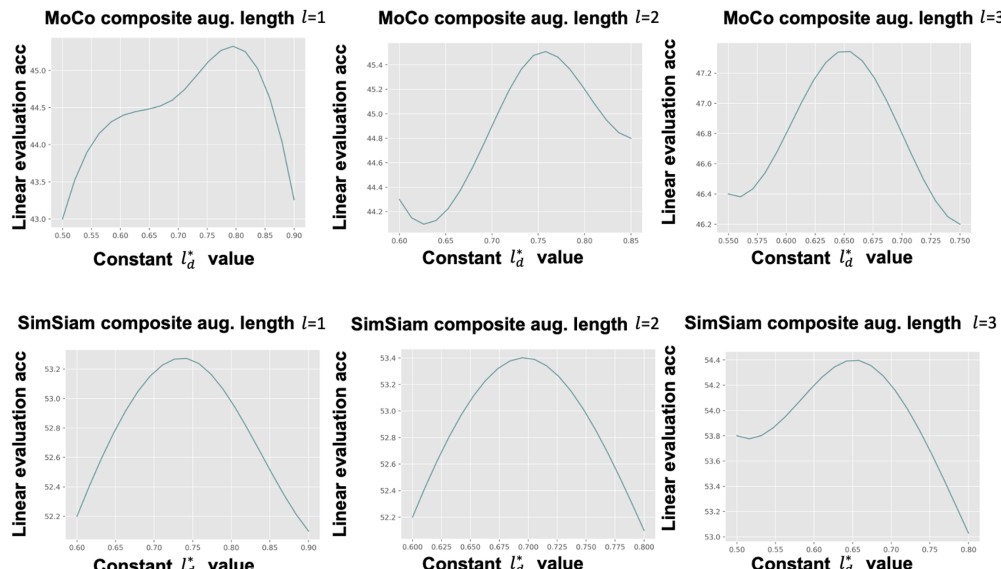

Figure 7: Linear performance accuracies of CoCor models trained with only one length of composite augmentation. CoCor with each length of composite augmentation is run with multiple different constant $l_d^*$ values.

## F  Ablation Study on CoCor without the MMNN

To evaluate the effect of MMNN, we run CoCor where $l_d^*$ is replaced by fixed constants. In practice, a constant is used for each length of composite augmentation in calculating the consistency loss. We run multiple experiments to select the constant for each length of composite augmentation. All models are ResNet-50 pre-trained on ImageNet-100 for 50 epochs. Figure 7 shows the linear evaluation results of CoCor running with one length of composite DA and with different constant $l_d^*$. For each length of composite augmentation, there exists an optimal constant which leads to the best result in linear evaluation. These

optimal constants are then used for the comparison with the MMNN. The best constants of MoCo and SimSiam are listed in Table 9.

| $\ell$ | 1 | 2 | 3 |
|---|---|---|---|
| MoCo | 0.80 | 0.75 | 0.65 |
| SimSiam | 0.75 | 0.70 | 0.60 |

Table 9: Constant $l_d^*$ used for the comparison with the MMNN.

## G   CoCor with much stronger data augmentations

CoCor is a systematic approach of leveraging the information provided by diverse augmentations. In order to further test the effect of composite augmentations of different strength, we run CoCor on MoCo and SimSiam with single augmentation length of 1, 2, 3, 5, and 10. We train ResNet-50 on ImageNet-100 for 50 epochs. Pre-trained backbones are evaluated on ImageNet-100 for linear classification. Table 10 shows the accuracies of evaluation.

| $\ell$ | 1 | 2 | 3 | 5 | 10 | baseline |
|---|---|---|---|---|---|---|
| MoCo | 47.90 | 47.70 | 48.08 | 45.16 | 43.07 | 42.66 |
| SimSiam | 54.01 | 54.20 | 55.16 | 53.50 | 52.70 | 44.60 |

Table 10: Linear evaluation of encoders trained by applying consistency loss on only one length $l$ of composite augmentations.

As it is shown in Table 10, introducing relatively weaker (shorter) data augmentation in pre-training leads to better performance on downstream tasks. When evaluating the pre-trained models on various downstream tasks, the test data for testing model performance are natural and realistic images. Thus, providing the encoder with less distorted views in pre-training can help the encoder better recognize natural images during inference.

It is also noteworthy that CoCor can leverage very strong augmentations of a length of 10 to improve the encoder's performance over baseline, which is even stronger than the data augmentations used in recent works that include strong augmentations such as CLSA Wang & Qi (2021) and RényiCL Lee & Shin (2023). Although it has been shown that very strong augmentations which can distort the identity of data too much may bring too much noise to the training process. The noise may result in performance degradation Chen et al. (2020a); Tian et al. (2020b), partial dimensional collapse, or even complete dimensional collapse Li et al. (2022a); Jing et al. (2021). However, CoCor is able to capture essential information from the highly distorted views and enrich the semantics of the feature space using these strong augmentations.

## H   CoCor's compatibility with more baseline contrastive learning methods.

To demonstrate CoCor's compatibility with additional baseline methods, e.g. BYOL Grill et al. (2020) and INTL Weng et al. (2023), we implement CoCor on these additional baseline methods. Linear evaluation results of these encoders across various datasets are presented in Table 11. These results indicate that CoCor not only enhances the generalizability of learned representations from contrastive methods like MoCo, SimSiam, and SupCon but also extends these benefits to other existing contrastive learning approaches.

## I   Training Time Comparison

CoCor applies the proposed consistency loss to views produced by various data augmentations. To address the concern of extra training time consumed by CoCor over its corresponding baseline methods, we compare the pre-training time and model performance in Table 12. Although CoCor requires more time per epoch,

Table 11: Top-1 accuracies (%) of linear evaluation. All ResNet-50 backbone encoders are pre-trained on ImageNet-100.

| Method | IN-100 | Cifar100 | CUB200 | Caltech101 | SUN397 | Food101 |
|---|---|---|---|---|---|---|
| BYOL Grill et al. (2020) | 68.05 | 60.13 | 20.27 | 78.90 | 39.03 | 53.01 |
| BYOL + CoCor (**Ours**) | **72.30** | **62.42** | **22.75** | **80.51** | **42.64** | **55.32** |
| INTL Weng et al. (2023) | 74.58 | 62.73 | 29.05 | 87.45 | 47.06 | 60.38 |
| INTL + CoCor (**Ours**) | **76.98** | **63.06** | **31.40** | **88.81** | **48.27** | **64.27** |

CoCor achieves better performance than its corresponding baselines with shorter training time. Additionally, CoCor takes similar amount of time to train as CLSA Wang & Qi (2021), which uses strong multi-view augmentations. However, CoCor achieves significantly better performance than CLSA with similar training time.

Table 12: Training Time Comparison on a machine with 4 A100 GPUs. Training time (in hour)/Top-1 linear evaluation results (in %) on ImageNet-1K is provided for comparison.

| Methods \ Epochs | 50 | 100 | 200 |
|---|---|---|---|
| MoCo He et al. (2020) | 13.4h/64.78 | 26.8h/66.81 | 53.6h/67.22 |
| SimSiam Chen & He (2021) | 13.2h/66.89 | 26.4h/68.12 | 52.8h/70.93 |
| CLSA Wang & Qi (2021) | 18.1h/67.95 | 36.2h/69.15 | 72.4h/71.19 |
| MoCo + CoCor | 18.5h/69.26 | 37.0h/70.76 | 74.0h/72.83 |
| SimSiam + CoCor | 18.1h/70.90 | 36.2h/72.03 | 72.4h/73.25 |

## J  Impact of the Data Augmentation Ordering in CoCor.

To evaluate the impact of the ordering of DAs in CoCor, we conducted experiments with various fixed DA orderings. We trained encoders using MoCo + CoCor with specific DA orderings. Results are shown in Table 13. For instance, in configurations CoCor-fixed 1/2, the order of DAs is predetermined. In a fixed order scenario, such as in CoCor-fixed 1, rotation is consistently applied before brightness whenever both augmentations are selected for an image. Results in Table 13 indicate that in CoCor, the ordering of DAs does not significantly affect the encoder's performance in downstream tasks. Thus, in final experiments, we use random DA ordering to explore a wider range of DA variation.

Table 13: Linear evaluation results on ImageNet-100 of CoCor models trained with different fixed DA orderings. Encoders are trained on ImageNet-100 for 200 epochs.

| | MoCo | CoCor - fixed 1 | CoCor - fixed 2 |
|---|---|---|---|
| Top-1 Acc | 67.04 | 71.32 | 71.47 |

