# OpenReview forum: "Contrastive Learning with Consistent Representations"
_TMLR — Accepted by TMLR_

### Review · Reviewer_e3Gj · 2024-07-03

**Summary Of Contributions:**

Data augmentation (DA) is important in contrastive learning for visual representation. Various kinds of data augmentation have been elaborately studied. However, how to efficiently compose different kinds of DA is rarely considered. In this paper, a measurement of DA consistency is proposed to make the view of stronger augmentation get smaller similarity with the original sample in the feature space, and the strength of the DA is estimated by an auxiliary network named monotonic mapping neural network (MMNN). Experimental results show that the proposed approach can be used as a plug-in module with different contrastive learning frameworks and achieves performance gain compared with them.

**Audience:**

Yes

**Broader Impact Concerns:**

No concern.

**Claims And Evidence:**

Yes

**Requested Changes:**

1. The MoCo, SimSiam, and SupCon are used in Table 1 to show the proposed DA consistent can be used together with different contrastive learning frameworks. All three frameworks are better compared in Table 2 to Table 7. In Table 2, as the SupCon is not compared, I would doubt that the DA consistently didn’t work.
2. It’s hard to understand the illustration of dimensional collapse evaluation in Fig. 4(b).
3. The strength of DA is measured with the length of DA in Section G in the appendix. However, it is measured by the monotonic mapping neural network. It’s not consistent.
4. Some numbers in the experiment section are non-corresponding, like 83.8 in Table 6 and 83.70 in Table 7.

**Strengths And Weaknesses:**

Strengths：
1. The motivation is clear that the view of stronger augmentation has a smaller similarity with the original sample, which is thoroughly introduced.
2. The experiments show the effectiveness of the proposed DA loss and the monotonic mapping neural network (MMNN).
3. The visualization in Fig. 4(a) verified the latent similarity is larger than the baseline when stronger DA is adopted.

Weaknesses：
1. The generalization of the proposed DA consistent is limited as only basic DA could be used, for example, color-related augmentations and affine transformations.
2. The teaser of Fig. 1(a) is overclaimed that the proposed approach cannot force the view of different DA to a regular structure, i.e., the circular ring. It’s better to explain the circular ring which indeed is the contour line of similarity.
3. Why is labeled data used for the monotonic mapping neural network? In my opinion, the consistency loss can be computed directly by ld(x; f, A1) - ld(x; f, A2), where A1 is stronger than A2 from the perspective of Fig. 3.

---

> ### Author Response · Authors · 2024-07-18
> **Response to Reviewer e3Gj**
>
> We appreciate reviewer e3Gj for their helpful and insightful comments and suggestions. Below we respond to the questions respectively.
>
> # ***W1***
> CoCor effectively manages a diverse array of DAs. As detailed in Appendix C, our implementation incorporates 14 different types of augmentations, which are all augmentation functions from PIL, a very popular Python image library. A special augmentation, RandomCrop, poses challenges as it introduces too much variance in creating views. However, our studies show that CoCor could handle multi-crop well if a separate MMNN is trained for each crop size, as illustrated by the following table.
>
> ***Table***: Top-1 linear evaluation accuracy comparison between methods incorporate multi-crop augmentations. ResNet-50 encoders trained on ImageNet-1K for 200 epochs are evaluated on ImageNet-1K.
>
> | Method | Top-1 Acc |
> |:----:|:----:|
> |SWAV*|72.67|
> |CLSA*|73.42|
> |CoCor*|75.06|
>
> *note all methods use multi-crop in their pre-training
>
> # ***W2***
> We appreciate reviewer e3Gj for their suggestion. We have refined Fig.1 and its caption in a revised version.
>
> # ***W3***
> Here, we aim to clarify the use of labeled data for the MMNN. We introduced the consistency loss (Equation(2)) to optimize the encoder towards DA consistency, guided by the MMNN's prediction of optimal latent similarity. However, the definition of optimality of the latent similarities in the encoder's latent space is measured by the encoder's performance in downstream visual recognition tasks. To this end, we train a linear classifier concurrently with the encoder on labeled data (Note that the encoder is frozen during the classifier update). As shown in Equation (3), the overall loss function for updating the encoder (${\theta_e}$) is influenced by the MMNN (${\theta_d}$), establishing a dependency denoted as ${\theta_e(\theta_d)}$. Thus, when calculating the classification loss $\text{CE}({\theta_e(\theta_d)})$ in Equation (4), instead of directly updating the encoder, we optimize ${\theta_d}$ to minimize this classification loss. A detailed derivation of this dependency and the bi-level optimization process is provided in Appendix B. ***In summary***, the MMNN learns the optimal latent similarity from labeled data to minimize classification loss, and the encoder learns this optimal latent similarity from the MMNN on unlabeled data for better consistency.

---

> > ### Author Response · Authors · 2024-07-18
> > **Requested changes**
> >
> > ***Table1***: Top-1 accuracy (%) comparison of linear evaluation. All encoders are ResNet-50 pre-trained on ImageNet-1K.
> > |Method|Epochs|IN-1K|Cifar10|Cifar100|CUB200|Caltech101|SUN397|Food101|Flowers102|Pets|
> > |:----|:----:|:----:|:----:|:----:|:----:|:----:|:----:|:----:|:----:|:----:|
> > |MoCo                              |200|67.56|92.24|74.33|41.54|92.00|58.28|69.84|88.86|81.74|
> > |MoCo+CoCor(***Ours***)|200|***72.83***|***93.28***|***76.19***|***44.11***|***93.10***|***60.15***|***71.10***|***90.39***|***83.08***|
> > |SimSiam                              |200|70.93|93.08|76.24|51.05|93.13|60.81|71.18|92.11|85.01|
> > |SimSiam+CoCor(***Ours***)|200|***73.25***|***94.04***|***78.12***|***54.50***|***94.51***|***63.09***|***73.31***|***92.84***|***87.24***|
> > |SupCon                              |100|74.18|93.17|76.55|52.30|93.28|61.03|71.54|91.34|85.35|
> > |SupCon+CoCor(***Ours***)|100|***76.24***|***94.61***|***77.91***|***55.73***|***94.66***|***63.80***|***74.09***|***92.17***|***87.21***|
> >
> > ***Table2***: Transfer learning results on VOC and COCO object detection tasks.
> >
> > | | | | VOC07+12| | | |COCO| | | |
> > |:----|:----:|:----:|:----:|:----:|:----:|:----:|:----:|:----:|:----:|:----:|
> > |Method|Epochs|AP|$AP_{50}$|$AP_{75}$|AP|$AP_{50}$|$AP_{75}$|$AP_{s}$|$AP_{m}$|$AP_{l}$|
> > |MoCo|200|50.23|76.52|54.42|34.23|55.08|36.60|16.67|37.27|50.97|
> > |MoCo+CoCor (***Ours***)|200|***51.11***|***77.41***|***54.88***|***39.13***|***58.30***|***42.19***|***23.19***|***43.45***|***52.13***|
> > |SimSiam|200|52.04|77.86|57.11|34.15|55.21|36.50|15.30|37.37|50.49|
> > |SimSiam+CoCor(***Ours***)|200|***54.31***|***80.60***|***59.58***|***34.80***|***56.29***|***37.05***|***17.53***|***39.88***|***51.74***|
> > |SupCon|100|47.41|75.83|53.17|34.26|55.11|36.35|14.34|37.40|50.56|
> > |SupCon+CoCor(***Ours***)|100|***49.03***|***77.22***|***54.19***|***35.67***|***56.10***|***37.98***|***16.63***|***38.71***|***51.29***|
> >
> > # ***C1***
> > Here we present additional comparisons of implementations based on SupCon. We pre-trained ResNet-50 encoders using both SupCon and SupCon-based CoCor on ImageNet-1K (IN-1K) for 100 epochs. The results of linear evaluation across various datasets, including IN-1K, are shown in Table1. Furthermore, we compare the performance of SupCon and SupCon+CoCor on object detection tasks, as shown in Table2. For a fair comparison, both methods utilize the same backbone architecture (ResNet-50) and are pre-trained on ImageNet-100 for 100 epochs. Enhanced performance is highlighted in the tables, demonstrating that CoCor effectively enhances the transferability and generalizability of representations learned by SupCon.
> >
> > # ***C2***
> > Dimensional collapse is a phenomenon where certain dimensions within the representation space of an encoder yield constant outputs for any input data. In such collapsed dimensions, data representations exhibit zero variance, making them poorly-separated and minimally-contributive to visual recognition tasks. To evaluate dimensional collapse in encoders, Principal Component Analysis (PCA) is often used. Specifically, for a dataset containing $N$ samples, the pre-trained encoder generates $N$ corresponding representations in the $\mathbb{R}^m$ representation space, forming an $N \times m$ matrix. PCA is applied to this matrix to extract $m$ singular values, ordered such that $\sigma_1 \geq \sigma_2 \geq \dots \geq \sigma_m$. Given that collapse typically correlates with smaller singular values, and PCA identifies orthogonal axes to maximize variance, it effectively detects dimensional collapse within an encoder. Figure 4(b) shows a comparison of the ordered singular values between baseline methods and CoCor, with CoCor exhibiting relatively larger singular values. This indicates that CoCor potentially reduces the issue of dimensional collapse in contrastive learning.

---

> ### Author Response · Authors · 2024-07-18
> **More requested changes**
>
> # ***C3***
> We introduce the Monotonic Mapping Neural Network (MMNN) due to the difficulty in knowing the optimal latent similarity for a given augmentation. Direct comparison of two different composite data augmentations (DAs) may be challenging; however, statistically, a composite augmentation consisting of 10 randomly-selected basic DAs is significantly stronger than one composed of a single random basic DA. In Appendix G, we demonstrate that CoCor can exploit data to which much stronger augmentations (length 10) is applied to achieve performance improvements over baseline methods. These findings indicate that CoCor effectively utilizes information from stronger augmentations compared to those used in CLSA[1] and RényiCL[2], which are known for their capability of leveraging strongly augmented views.
>
> # ***C4***
> We thank e3Gj for pointing out this issue. In revising our submission, we reproduced several experiments, leading to variations in our results. We apologize for any confusion this may have caused. We have ensured that the results are now consistent throughout the revised manuscript.
>
> [1]Xiao Wang and Guo-Jun Qi. Contrastive learning with stronger augmentations. arXiv preprint arXiv:2104.07713, 2021.
>
> [2]Kyungmin Lee and Jinwoo Shin. Renyicl: Contrastive representation learning with skew renyi divergence, 2023.

---

> > ### Comment · Reviewer_e3Gj · 2024-08-10
> >
> > Thanks for the detailed feedback, which solves most of my concerns.  However, the response to the generalization of the DA consistency is still not satisfied.  Firstly, a separate MMNN is trained for the RandomCrop in the response, which will burden the model when various kinds of DA are used. Second, it's better to give a guideline for when a separate MMNN should be used.

---

> > > ### Author Response · Authors · 2024-08-10
> > > **Response to reviewer e3Gj**
> > >
> > > Dear reviewer e3Gj,
> > >
> > > Thank you very much for your feedback. We would like to respond to your concerns as follows.
> > >
> > > 1) Although an MMNN is required for each sized RandomCrop, for the MMNN is lightweight (a 3-layer MLP), we did not observe a significant increase in training time when compared to existing contrastive approaches that are able to utilize multi-crop views, such as CLSA [1]. Table 12 in our revised manuscript provides a comparison of training times for a single-sized crop.
> > >
> > > 2) We appreciate your suggestion to include more detailed guidelines for the use of the MMNN. We will incorporate this in our revised manuscript.
> > >
> > > Thank you once again for your time and helpful comments.
> > >
> > > Best regards,
> > >
> > > The authors
> > >
> > > [1] Xiao Wang and Guo-Jun Qi. Contrastive learning with stronger augmentations. arXiv preprint arXiv:2104.07713, 2021.

---

### Review · Reviewer_q2iP · 2024-07-04

**Summary Of Contributions:**

This paper focuses on investigating the data augmentation (DA) of contrastive learning (typically the self-supervised learning for vision data). The motivation is that the intricacies of data augmentation and representation learning may lead to a performance degradation if augmentation functions are not judiciously chosen. This paper proposes Contrastive Learning with Consistent Representations (CoCor), including introducing the set of composite augmentations, defining the DA consistency. It further proposes to learn the optimal mapping locations as a function of DA, all while preserving a desired monotonic property relative to DA intensity. Experimental results demonstrate that effectiveness of CoCor in enhancing the generalizability and transferability of learned representations comparing to baseline methods

**Audience:**

Yes

**Claims And Evidence:**

No

**Requested Changes:**

My main concern is the practicality of the proposed method, it seems the proposed method is complicated in computation, and this paper does not provide strong results in experiments (see weaknesses) . It is better to add more experimental results to support the claims or reword the claims.

**Strengths And Weaknesses:**

**Strengths:**

The motivation of the proposed method is clear. The proposed method intuitively can improve the performance of SSL methods, if it is well designed to exploit the magnitude (of consistency) of data augmentation. The introduce of data augmentation consistency is new to me.
This paper is overall well organized, the description is also clear.
The experimental results of “Composite augmentation length in pre-training” is interesting.


**Weaknesses:**

1.	This paper should provide the computation cost of the CoCor. E.g., The overall time cost to train baselines+CoCor, compared to the baselines. Based on the description of CoCor, I feel the proposed method of this paper will introduce significant additional computation cost, due to the bilevel-optimization. I doubt the practicability of the proposed method.
2.	This paper claims that “ CoCor achieves state-of-the-art results for various downstream tasks.”. I donot think the current experimental results can support this claim. The experimental results currently are weak:
 (1) In Section 5.2 ‘Main results”, Why not show the results of linear evaluation on ImageNet-1K itself?, like the experimental setup in Simsiam paper (Chen&He 2021). It is important to show this result;
(2) In Table 3, the Simsiam method is significantly lower the Simsiam paper reported, e..g, Simsiam obtains 57.0 AP on VOC07+12  and 39.2 AP on COCO detection reported in Simsiam paper (Chen&He 2021) while 50.23 AP and 34.23 AP only in this paper. I think this paper should clarify it.
(3) I donot think MoCO and SimSiam is currently the state-of-the-art SSL baselines, there are many better methods, e..g, Barlow Twins[1], INTL[2] (INTL [2] can obtains 75.2 accuracy of linear evaluation on ImageNet using 200epoch training with strong data argumentation) and 40.7 AP on COCO). This paper should conduct experiments to compare these methods.
3.	This paper claims that “Moreover, it can be readily integrated into existing contrastive learning frameworks, effectively imposing DA consistency on the encoder”. I think this paper should conduct more experiments to support this claim, since the experiments are only based on Moco and SimSiam. This paper should conduct more experiments on SSL method using regularization, e.g., Barlow Twins[1], VICReg[3], and INTL [2]


4.  This paper claims “Our studies show that the ordering of the basic augmentations in a composite augmentation does not have a significant impact on performance.”. I think this paper should provide some evidence (in Appendix) to support this claim. E.g., how this paper gets the observations.,

5.   Some notation is not well explained (If I am correct), e.g., (1) What is $N_a$ in the last row of page 4? (2) what is $N_0$ in the Definition 2?

**Ref:**

[1] Barlow Twins: Self-Supervised Learning via Redundancy Reduction, ICML 2021
[2]Vicreg: Variance-invariance-covariance regularization for self-supervised learning, ICLR 2022
[3 ] MODULATE YOUR SPECTRUM IN SELF-SUPERVISED LEARNING, ICLR, 2024

---

> ### Author Response · Authors · 2024-07-18
> **Response to reviewer q2iP**
>
> We appreciate reviewer q2iP for their helpful and insightful comments and suggestions. Below we respond to the questions respectively.
>
> ***Table1:*** Training Time Comparison on a machine with 4 A100 GPUs. Training time (in hour)/Top-1 linear evaluation results (in \%) on ImageNet-1K is provided for comparison.
> |Method\Epochs|50|100|200|
> |:----|:----:|:----:|:----:|
> |MoCo            | 13.4h / 64.78 | 26.8h / 66.81 | 53.6h / 67.22  |
> |SimSiam       | 13.2h / 66.89 | 26.4h / 68.12 | 52.8h / 70.93  |
> |CLSA[1]            | 18.1h / 67.95 | 36.2h / 69.15 | 72.4h / 71.19  |
> |MoCo + CoCor     | 18.5h / 69.26 | 37.0h / 70.76 | 74.0h / 72.83  |
> |SimSiam + CoCor | 18.1h / 70.90 | 36.2h / 72.03 | 72.4h / 73.25 |
> # ***W1***
> We provide the training time and linear accuracy comparison in Table1. Although CoCor requires more time per epoch, CoCor achieves better performance than its corresponding baselines with shorter training time. Additionally, CoCor takes similar amount of time to train as CLSA[1], which uses strong multi-view augmentations. However, CoCor achieves significantly better performance than CLSA[1] with similar training time.
>
> ***Table2***: Top-1 accuracy (%) comparison of linear evaluation. All encoders are ResNet-50 pre-trained on ***ImageNet-1K***.
> |Method|Epochs|IN-1K|Cifar10|Cifar100|CUB200|Caltech101|SUN397|Food101|Flowers102|Pets|
> |:----|:----:|:----:|:----:|:----:|:----:|:----:|:----:|:----:|:----:|:----:|
> |MoCo                              |200|67.56|92.24|74.33|41.54|92.00|58.28|69.84|88.86|81.74|
> |MoCo+CoCor(***Ours***)|200|***72.83***|***93.28***|***76.19***|***44.11***|***93.10***|***60.15***|***71.10***|***90.39***|***83.08***|
> |SimSiam                              |200|70.93|93.08|76.24|51.05|93.13|60.81|71.18|92.11|85.01|
> |SimSiam+CoCor(***Ours***)|200|***73.25***|***94.04***|***78.12***|***54.50***|***94.51***|***63.09***|***73.31***|***92.84***|***87.24***|
> |SupCon                              |100|74.18|93.17|76.55|52.30|93.28|61.03|71.54|91.34|85.35|
> |SupCon+CoCor(***Ours***)|100|***76.24***|***94.61***|***77.91***|***55.73***|***94.66***|***63.80***|***74.09***|***92.17***|***87.21***|
>
>
> ***Table3***: Top-1 accuracy (%) comparison of linear evaluation. All encoders are ResNet-50 pre-trained on ***ImageNet-100***.
> |Method|Epochs|IN-100|Cifar10|Cifar100|CUB200|Caltech101|SUN397|Food101|Flowers102|Pets|
> |:----|:----:|:----:|:----:|:----:|:----:|:----:|:----:|:----:|:----:|:----:|
> |MoCo                              |200|67.22|82.15|59.17|21.52|79.23|39.43|52.97|71.77|56.94|
> |MoCo+CoCor(***Ours***)|200|***71.66***|***83.87***|***59.64***|***22.57***|***81.51***|***41.95***|***54.81***|***75.65***|***60.23***|
> |SimSiam                              |200|73.88|84.66|61.82|26.13|85.11|45.96|58.73|82.40|65.60|
> |SimSiam+CoCor(***Ours***)|200|***83.74***|***86.89***|***66.36***|***34.50***|***87.85***|***49.92***|***62.48***|***87.95***|***76.04***|
> |SupCon                              |100|80.16|84.30|61.33|26.46|85.04|41.95|52.45|76.99|73.75|
> |SupCon+CoCor(***Ours***)|100|***82.14***|***85.49***|***62.18***|***27.40***|***87.12***|***42.87***|***53.93***|***79.04***|***75.95***|
>
> # ***W2***
> (1) We further provide ImageNet-1K and ImageNet-100 pre-trained encoders' linear evaluation results on ImageNet-1K and ImageNet-100 in Table2 and Table3, respectively. (2) In the SimSiam paper, an encoder pre-trained on ImageNet-1K for 200 epochs is used for object detection tasks. However, we utilize encoder pre-trained on ImageNet-100 for 200 epochs for object detection. Despite the difference in performance, we ensure all comparisons are made under the same experiment settings in our submission for the sake of fairness. (3) To demonstrate CoCor's compatibility with additional baseline methods, e.g. BYOL[2] and INTL[3], we pre-trained encoders on ImageNet-100 for 200 epochs. Linear evaluation results of these encoders across various datasets are presented in Table4. These results indicate that CoCor not only enhances the generalizability of learned representations from contrastive methods like MoCo, SimSiam, and SupCon but also extends these benefits to other models.
>
> ***Table4***: Top-1 accuracies (\%) of linear evaluation. All encoders are ResNet-50 pre-trained on ImageNet-100 for 200 epochs.
> |Method|IN-100|Cifar100|CUB200|Caltech101|SUN397|Food101|
> |:----|:----:|:----:|:----:|:----:|:----:|:----:|
> |BYOL[2]|68.05|60.13|20.27|78.90|39.03|53.01|
> |BYOL+CoCor(***Ours***)|***72.30***|***62.42***|***22.75***|***80.51***|***42.64***|***55.32***|
> |INTL[3]|74.58|62.73|29.05|87.45|47.06|60.38|
> INTL+CoCor(***Ours***)|***76.98***|***63.06***|***31.40***|***88.81***|***48.27***|***64.27***|

---

> > ### Author Response · Authors · 2024-07-18
> > **more response**
> >
> > # ***W3***
> > We conduct experiments regarding CoCor's compatibility with more contrastive learning methods. Details can be found in our response at ***W2(3)*** and ***Table4***.
> >
> > ***Table5***: Linear evaluation results on ImageNet-100 of CoCor models trained with different fixed DA orderings. Encoders are trained on ImageNet-100 for 200 epochs.
> > | | MoCo|CoCor-fixed 1|CoCor-fixed 2|
> > |:----:|:----:|:----:|:----:|
> > |Top-1 Acc| 67.04|71.32|71.47|
> >
> > # ***W4***
> > To evaluate the impact of the ordering of DAs in CoCor, we conducted experiments with various fixed DA orderings. We trained encoders using MoCo + CoCor with specific DA orderings. Results are shown in Table5. For instance, in configurations CoCor-fixed 1/2, the order of DAs is predetermined. In a fixed order scenario, such as in CoCor-fixed 1, rotation is consistently applied before brightness whenever both augmentations are selected for an image. Results in Table5 indicate that in CoCor, the ordering of DAs does not significantly affect the encoder's performance in downstream tasks. Thus, in final experiments, we use random DA ordering to explore a wider range of DA variation.
> >
> > # ***W5***
> > We appreciate reviewer q2iP for the comments. Here we hope to clarify the notations (1) As explained in Definition 1, $N_a$ is the number of basic DAs in the DA set. (2) $\mathbb{N}_0$ denotes the set of natural numbers. We improved the clarity addressing the concerns raised in a revised manuscript.
> >
> > [1]Xiao Wang and Guo-Jun Qi. Contrastive learning with stronger augmentations. arXiv preprint arXiv:2104.07713, 2021.
> >
> > [2]Grill, Jean-Bastien, Florian Strub, Florent Altché, Corentin Tallec, Pierre Richemond, Elena Buchatskaya, Carl Doersch et al. "Bootstrap your own latent-a new approach to self-supervised learning." Advances in neural information processing systems 33 (2020): 21271-21284.
> >
> > [3]Weng, Xi, Yunhao Ni, Tengwei Song, Jie Luo, Rao Muhammad Anwer, Salman Khan, Fahad Khan, and Lei Huang. "Modulate Your Spectrum in Self-Supervised Learning." In The Twelfth International Conference on Learning Representations.

---

### Review · Reviewer_KPsZ · 2024-07-04

**Summary Of Contributions:**

The paper presents an approach to addressing a key challenge in contrastive learning: the dependency on the quality of manually chosen data augmentation functions. Recognizing the critical role of data augmentations in providing informative views without explicit labels, the authors introduce Contrastive Learning with Consistent Representations (CoCor). Central to CoCor is the Data Augmentation consistency, which ensures optimal mapping of augmented input data in the representation space, aligned with the intensity of the applied augmentation. The proposed method systematically incorporates diverse data augmentations and learns optimal mapping locations while preserving a monotonic relationship with DA intensity. The experimental results are noteworthy, demonstrating that CoCor significantly enhances the generalizability and transferability of learned representations compared to baseline methods. This approach not only addresses the complexities inherent in data augmentations and representation learning but also offers a systematic solution that could be highly beneficial for future research in the field.

**Audience:**

Yes

**Broader Impact Concerns:**

No need to discuss broader impact concerns.

**Claims And Evidence:**

No

**Requested Changes:**

1. Discussion on the Difference with Reference[1].

2. The authors should address the reasons for any degraded results observed. It would be beneficial to explain why the proposed method might underperform in certain scenarios and attempt to replicate the results under the same settings and baseline conditions. This would provide a clearer understanding of the method's strengths and weaknesses and offer insights into potential areas for improvement.

3. The authors should strive to improve the clarity and readability of the paper. Simplifying complex sentences and ensuring proper punctuation would significantly enhance comprehension.

**Strengths And Weaknesses:**

Pros:
1. The idea of incorporating data augmentations into the loss function and making the encoder aware of the augmentations is commendable. Unlike AugSelf, this paper delves deeply into designing a metric to rank the augmentations, which is a novel and valuable contribution.


Cons:
1. Writing Clarity: The writing of this paper is somewhat confusing. For example, it begins by introducing augmentation in a supervised setting, but this is not connected to the rest of the paper. The sentences in the introduction are somewhat disjointed and the authors tend to use very long and complex sentences. Missing commas exacerbate this issue, making the text difficult to understand. For instance, in Definition 3, the lack of a comma after "A" made it challenging to understand what "A" represents.

2. Formulation. The formulation presented in the paper seems a bit confusing, and the star representing the optimal should not be on the $l_d$, but on the parameter $\theta_d$.

3. Literature Review: The paper misses a very relevant piece of literature. The formulation (learning another neural network as a new metric) and the idea of the proposed method are quite similar to [1], which focuses on crop size as a data augmentation method. This paper, however, extends the focus to general data augmentations. Including and discussing this related work would provide a more comprehensive context for the contributions of the current paper.

4. The justification of stronger augmentations. Simply taking more operations as stronger augmentation is a bit unreasonable, and at least from my humble view, the intensity should be a key value. For example, conducting twice saturations with smaller parameters should be less than one saturation operation with a larger value.

5. The experiments. I found the results in all the tables are much lower than the MoCov2 and SimSiam are much lower than the original paper and results reported by [1] and Augself.


[1] T. Zhang C. Qiu, W. Ke, S. Süsstrunk, and M. Salzmann, "Leverage Your Local and Global Representations: A New Self-Supervised Learning Strategy" CVPR 2022

---

> ### Author Response · Authors · 2024-07-18
> **Response to reviewer KPsZ**
>
> We appreciate reviewer KPsZ for their helpful and insightful comments and suggestions. Below we respond to the questions respectively.
>
> # ***W1***
> We appreciate the helpful comments from reviewer KPsZ on our paper. The reason we introduce data augmentations (DAs) under supervised settings is motivated by evidence showing that stronger DAs are effective in supervised learning, yet their potential in contrastive learning is not well explored. Therefore, we included a brief discussion on the effectiveness of DAs in supervised settings. We also thank the reviewer for pointing out the missing commas; these have been corrected in a revised manuscript.
>
> # ***W2***
> We thank reviewer for their comments on the formulation. We would like to clarify the optimality regarding $l_d$. The concept of $l_d^*$ is introduced in Definition 4, where we measure the consistency level without involving the MMNN ($\theta_d$). Because identifying $l_d^*$ is challenging, we subsequently introduce the MMNN and the bi-level optimization in following sections.
>
> # ***W3***
> Here we hope to discuss the related work "Leverage Your Local and Global Representations: A New Self-Supervised Learning Strategy" CVPR 2022 (LoGo)[1]. Instead of clustering all views together, LoGo considers the differences between local (extracting smaller regions from images) and global (extracting larger patches) crops. LoGo proposes to use a trainable discriminator network to model the variance between local patches and train the encoder to learn the differences between patches learned by the discriminator. However, CoCor considers more diverse and a larger range of DA strength for contrastive learning. Additionally, instead of learning to discriminate if two views are from the same image, CoCor trains the MMNN to directly learn the optimal latent similarity between augmented and original images given the DA composition. We added this discussion in a revised version of our manuscript.
>
> # ***W4***
> Similar to CLSA[2], we employ a fixed intensity for each basic DA. Therefore, applying each augmentation more times results in stronger distortion in a given image. The development of a MMNN that accounts for the intensity of each sampled DA is an interesting direction for future exploration.
>
> # ***W5***
> We reproduced all the experiments of existing methods. The reason for the differences between results from our experiments and the ones reported in the original literature is because of the experiment settings. Specifically, in our manuscript, Table1 and Table3 report the linear evaluation and objective detection results of encoders pre-trained on ImageNet-100 for 200 epochs. And Table2 evaluates encoder pre-trained on ImageNet-1K for 200 epochs. However, the objective detection results in MoCoV2 and SimSiam papers are from encoders pre-trained on ImageNet-1K for 200/800 epochs. In AugSelf, all results are from encoders pre-trained on ImageNet-100 for 500 epochs. Although the results from our reproduced experiments differ from those in the original literature, all comparisons in our submission are made under the same experiment settings for the sake of fairness.
>
> [1]Zhang, Tong, Congpei Qiu, Wei Ke, Sabine Süsstrunk, and Mathieu Salzmann. "Leverage your local and global representations: A new self-supervised learning strategy." In Proceedings of the IEEE/CVF conference on computer vision and pattern recognition, pp. 16580-16589. 2022.
>
> [2]Xiao Wang and Guo-Jun Qi. Contrastive learning with stronger augmentations. arXiv preprint arXiv:2104.07713, 2021.

---

> ### Author Response · Authors · 2024-07-18
> **Requested changes**
>
> # ***C1***
> We discussed this related work above in ***W3***. We also added this discussion to our revised manuscript.
>
> ***Table1***: Performance comparison between MoCo and CoCor using only affine DAs. Encoders are pre-trained on ImageNet-100 for 200 epochs.
> | | ImageNet-100 | Caltech101| Pets|
> |:----|:----:|:----:|:----:|
> |MoCo|81.1|93.2|85.4|
> |MoCo+CoCor(affine)|79.3|91.9|84.7|
>
> # ***C2***
> Here we present a scenario in which the encoder pre-trained with MoCo+CoCor underperforms its baseline, MoCo. In Table1, we show the linear evaluation results of MoCo and MoCo+CoCor (affine), where only affine transformations are used for pre-training. Learning variations introduced by specific augmentations may decrease the encoder's performance on certain datasets. These data augmentations (DAs) can generate data that is 'out-of-distribution' for the target dataset. For instance, rotation produces rotated dog images on the pre-training dataset, while the target dataset contains only non-rotated dog image. This mismatch in data distribution can consequently degrade performance.
>
> # ***C3***
> We appreciate the suggestions provided by Reviewer KPsZ. In response, we have enhanced the clarity of our manuscript by addressing the concerns raised.

---

> > ### Comment · Reviewer_KPsZ · 2024-07-28
> > **response to the rebuttal**
> >
> > I would like to thank the reviewers for their constructive feedback.
> >
> > I have re-examined the references and found discrepancies in the pre-training details of the MoCo v2 model. Specifically, reference [1] pre-trains the MoCo v2 model for 200 epochs, while reference [2] uses 500 epochs. More importantly, the baseline performance of the MoCo v2 method varies across different datasets when compared to both reference [1] and Aug-Self. For example, in the Caltech dataset, the MoCo v2 baseline performance reported in our paper (79.23) is higher than that in reference [1] (74.12) and Aug-Self (77.25). However, in other datasets, the baseline performance in the paper is notably lower than the figures presented in reference [1] and Aug-Self. This inconsistency raises concerns about the reported results.
> >
> > Regarding the data augmentation process, could you clarify whether you are applying different random parameters to the augmentations of the same image within the current epoch, or if you are keeping the parameters consistent across all images?

---

> > > ### Author Response · Authors · 2024-07-28
> > > **Response to reviewer KPsZ**
> > >
> > > Dear Reviewer KPsZ,
> > >
> > > Thank you very much for your feedback. We would like to address your concerns as follows:
> > >
> > > We reimplemented all existing methods using publicly available code. However, for some methods, such as [1] and [2], the code is not publicly available. In these cases, we adopted the reported results in their paper and ran other methods under the experiment settings detailed in their papers. To enhance clarity, we added a brief description of the pre-training settings to each table caption in the revised manuscript (marked in blue).
> > >
> > > The reason our results are lower than those in AugSelf[3] is that AugSelf pre-trains encoders for 500 epochs on ImageNet-100, while we use 200-epoch pre-trained encoders. Additionally, we believe the higher performance in Caltech101 is due to different dataset splits between our implementation and the original. Despite these differences, all comparisons in our manuscript were conducted under the same experimental settings to ensure fairness. We will also add notes to our tables to clarify whether they are reproduced by us or adopted from the original paper in the next revision.
> > >
> > > Similar to CLSA, we employed a fixed intensity for each basic data augmentation as explained in our previous response to Reviewer W4.
> > >
> > > Thank you,
> > >
> > > The Authors
> > >
> > > [1] Zhang, Junbo, and Kaisheng Ma. "Rethinking the augmentation module in contrastive learning: Learning hierarchical augmentation invariance with expanded views." Proceedings of the IEEE/CVF Conference on Computer Vision and Pattern Recognition. 2022.
> > >
> > > [2]Tete Xiao, Xiaolong Wang, Alexei A Efros, and Trevor Darrell. What should not be contrastive in contrastive learning. In International Conference on Learning Representations, 2021.
> > >
> > > [3]Lee, Hankook, et al. "Improving transferability of representations via augmentation-aware self-supervision." Advances in Neural Information Processing Systems 34 (2021): 17710-17722.

---

> > > > ### Comment · Reviewer_KPsZ · 2024-08-09
> > > > **Response to your rebuttal**
> > > >
> > > > Could I confirm whether you apply the your augmentation strategy to the MoCov2 baseline, where each augmentation has a fixed parameter?

---

> > > > > ### Author Response · Authors · 2024-08-10
> > > > > **Response to reviewer KPsZ**
> > > > >
> > > > > Dear Reviewer KPsZ,
> > > > >
> > > > > Thank you for your question. We would like to clarify the augmentation strategy applied in implementing CoCor based on baseline contrastive methods such as MoCo, SimSiam, BYOL, SupCon, and INTL.
> > > > >
> > > > > Using MoCoV2-based CoCor as an example, as outlined in Equation (3), our method includes two loss terms: contrastive loss and consistency loss. For the **contrastive loss**, we follow the augmentation strategy detailed in the original MoCoV2 paper [1] (which uses random parameter DAs) to generate the two views of each image. For the **consistency loss**, however, we employ parameter-fixed basic augmentations to compose strong DAs, generate image views, and compute the consistency loss.
> > > > >
> > > > > Thank you again for your time and valuable comments.
> > > > >
> > > > > Best regards,
> > > > >
> > > > > The Authors
> > > > >
> > > > > [1] Xinlei Chen, Haoqi Fan, Ross Girshick, and Kaiming He. Improved baselines with momentum contrastive learning. arXiv preprint arXiv:2003.04297, 2020b.

---

### Decision · Action_Editor_tGUN · 2024-08-23

**Recommendation:** Accept as is

**Comment:**

Despite expressing some concerns in their final evaluation, including w.r.t. the generality/uniformity of the approach across different augmentations, some reported baseline numbers, and the computational cost of the method, the reviewers all acknowledge that the paper introduces some interesting ideas.

**Audience:**

Yes

**Claims And Evidence:**

Mostly. Reviewer KPsZ nonetheless expresses some concerns about some of the reported baseline numbers. However, considering that the experiments were all run under the same settings, as argued by the authors, the AE believes that the evidence is sufficient.